# TIM4 expression by dendritic cells mediates uptake of tumor-associated antigens and anti-tumor responses

Nicoletta Caronni[1,2,18✉], Giulia Maria Piperno[1,18], Francesca Simoncello[1], Oriana Romano [3], Simone Vodret[4], Yuichi Yanagihashi[5], Regine Dress [6], Charles-Antoine Dutertre[6], Mattia Bugatti[7], Pierre Bourdeley[8], Annalisa Del Prete [9,10], Tiziana Schioppa [9,10], Emilia Maria Cristina Mazza [3,11], Licio Collavin[12], Serena Zacchigna [4,13], Renato Ostuni [2,14], Pierre Guermonprez [8], William Vermi [7,15], Florent Ginhoux [6,16,17], Silvio Bicciato [3], Shigekatzu Nagata [5] & Federica Benvenuti[1✉]

Acquisition of cell-associated tumor antigens by type 1 dendritic cells (cDC1) is essential to induce and sustain tumor specific CD8[+] T cells via cross-presentation. Here we show that capture and engulfment of cell associated antigens by tissue resident lung cDC1 is inhibited during progression of mouse lung tumors. Mechanistically, loss of phagocytosis is linked to tumor-mediated downregulation of the phosphatidylserine receptor TIM4, that is highly expressed in normal lung resident cDC1. TIM4 receptor blockade and conditional cDC1 deletion impair activation of tumor specific CD8[+] T cells and promote tumor progression. In human lung adenocarcinomas, TIM4 transcripts increase the prognostic value of a cDC1 signature and predict responses to PD-1 treatment. Thus, TIM4 on lung resident cDC1 contributes to immune surveillance and its expression is suppressed in advanced tumors.

[1] Department of Cellular Immunology, International Centre for Genetic Engineering and Biotechnology, ICGEB, Trieste, Italy. [2] San Raffaele Telethon Institute for Gene Therapy (SR-TIGET), IRCCS San Raffaele Scientific Institute, Milan, Italy. [3] Department of Life Sciences, University of Modena and Reggio Emilia, Modena, Italy. [4] Cardiovascular Biology Laboratory, International Centre for Genetic Engineering and Biotechnology (ICGEB), Trieste, Italy. [5] Laboratory of Biochemistry & Immunology, World Premier International Research Center, Immunology Frontier Research Center, Osaka University, Suita, Osaka, Japan. [6] Singapore Immunology Network (SIgN), Agency for Science, Technology and Research (A*STAR), Singapore, Singapore. [7] Department of Molecular and Translational Medicine, School of Medicine, University of Brescia, Brescia, Italy. [8] Centre for Inflammation Biology and Cancer Immunology, School of Immunology and Microbial Sciences, Faculty of Life Sciences and Medicine, King's College London, London, UK. [9] Department of Molecular and Translational Medicine, University of Brescia, Brescia, Italy. [10] Humanitas Clinical and Research Center-IRCCS, Rozzano-Milano, Italy. [11] Laboratory of Translational Immunology, Humanitas Clinical and Research Center-IRCCS, Rozzano-Milano, Italy. [12] Department of Life Sciences (DSV), University of Trieste, Trieste, Italy. [13] Department of Medical, Surgical and Health Sciences, University of Trieste, Trieste, Italy. [14] Vita-Salute San Raffaele University, Milan, Italy. [15] Department of Pathology and Immunology, Washington University School of Medicine, St Louis, St. Louis, MO, USA. [16] Shanghai Institute of Immunology, Shanghai JiaoTong University School of Medicine, Shanghai, China. [17] Translational Immunology Institute, SingHealth Duke-NUS Academic Medical Centre, Singapore, Singapore. [18]These authors contributed equally: Nicoletta Caronni, Giulia Maria Piperno. ✉email: caronni.nicoletta@hrs.it; benvenuti@icgeb.org

Tissue resident Batf3-dependent type 1 dendritic cells (DCs), named cDC1, excel in activation of CD8[+] T cells to exogenously acquired antigens, and are essential to detect tumors and to mediate rejection of nascent lesions[1–4]. Moreover, intratumoral cDC1 support the function of T cells during T-cell mediated immune clearance[5,6], and their abundance in several human cancers correlates with better prognosis[5,7,8]. cDC1 accumulation in tumors depends on NK cells and CCL5[8,9] and cause, in turn, recruitment and expansion of tumor specific T cells in situ[7,10]. These positive correlations likely reflect residual cDC1 activity, as many reports suggest that DCs are negatively affected by signals present in the tumor immune microenvironment (TME)[10–18].

Previous work established that cDC1-mediated anti-tumor immunity primarily relies on engulfment of dying tumor cells to acquire cell-associated antigens[5,6,16]. Still, the receptors that control acquisition of tumor cell-associated antigens by cDC1 in tumor tissues, and whether this process is targeted by the suppressive TME has not been investigated. Putative receptors such as CD36, DEC-205, DNGR1/Clec9A, SCARF1 and AXL control recognition/binding of dying cells and cross-presentation of cell associated antigens by cDC1[19–24], in various contexts. TIM4 (T-cell immunoglobulin mucin-4), that recognizes phosphatidylserine (PtdSer) on apoptotic cells (AC), is predominantly expressed on resident peritoneal macrophages[25–27] and some subsets of DCs[28–32]. The expression of these phagocytic receptors on tissue resident cDC1, their contribution to tumor antigen uptake and modulation during tumor growth remains however poorly explored.

In this study, we use a transplantable orthotopic KP model of adenocarcinoma of the lung (derived from Kras/p53 KP mice[16,33,34]) to study the ability of lung resident cDC1 to engulf cell-associated antigens during tumor development. cDC1 actively engulf and present cell-associated antigens to CD8[+] T cells at early stages of tumor development, yet antigen uptake, presentation and T cell activation is inhibited in advanced tumors. Gene expression profiling show high expression of TIM4 in normal lungs cDC1 and its downregulation in cells associated with late tumors. In addition, profiling across lung phagocytes indicates that, unlike what has been described in other tissues, TIM4 is not expressed on lung resident macrophages or macrophages associated to lung tumors. Receptor blockade and cDC1 genetic deletion of TIM4 regulates the capacity of cDC1 to engulf dying cells and tumor cells in the lung, and to induce cancer specific CD8 responses that control tumor growth. In human lung adenocarcinomas a combined cDC1/Timd4 gene signature predicts survival in early stages patients, and responses to PD-1 treatment in a cohort of lung adenocarcinoma patients.

## Results

**cDC1 fail to uptake cell associated antigens in late lung tumors tissue.** To explore uptake of cell associated antigens by lung resident cDC1 during tumor development, we established orthotopic tumors by intravenous inoculation (i.v.) of Kras[G12D/+];p53[−/−] primary tumor cells (KP)[33,34]. We defined two time points for analysis, early tumors (KP(e), days 10–14) when the tumor area occupies between 5–10% of the lung parenchyma and a late time point, when nodules occupy more than 30% of the total lung area (KP (l), days 28–35) (Supplementary Fig. 1a). Mice injected with PBS were used as control (normal lungs, nLung). Established tumors showed an increased frequency of neutrophils, unchanged frequencies of SiglecF[+] CD11c[+] alveolar macrophages (AM) and Ly6C monocytes and decreased frequencies of T cells (Supplementary Fig. 1b). The DCs compartment (Supplementary Fig. S1c–e) in late tumors was characterized by an increment of mo-DCs and cDC2 over cDC1, in line with previous data[16,18]. Expression of cDC1 lineage specific and maturation markers was not affected in tumor-associated cDC1 (Supplementary

Fig. 1f). To examine the ability of cDC1 to engulf dying cells, we first administered labeled apoptotic thymocytes intratracheally (i.t.) to control or tumor bearing animals. Quantification of fluorescence associated with phagocytes 2 h after inoculation showed a high uptake by AM and a lower uptake by cDC1 and cDC2 (Figs. 1a and S2a). Uptake by cDC2 and AM remained unchanged in early and late tumors as compared to controls, whereas uptake by cDC1 was significantly inhibited in late tumors (Fig. 1a). To directly examine the uptake of cellular tumor fragments in tissues, we generated a KP-BFP reporter line. At early time points after tumor inoculation we observed a higher BFP fluorescence in lung cDC1 than in cDC2 and AM, likely due to faster degradation of endocytic cargo in the latter subsets (Supplementary Fig. 2b)[5,35]. In more advanced tumors, BFP associated with cDC1 was significantly decreased whereas it remained unchanged in cDC2 and AM, suggesting a selective reduction in the capacity to engulf tumor cells by cDC1 (Fig. 1b and Supplementary Fig. 2b). To visualize the uptake at the singe cell level we next isolated cDC1 from control, early, and late KP tumor-bearing mice and incubated them ex-vivo with labeled apoptotic thymocytes. cDC1 from normal lung and early tumors captured and engulfed dying cells efficiently. In contrast, cDC1 isolated from late tumors were impaired in binding to and phagocytosis of dying cells (Fig. 1c, d and Supplementary Fig. 2c, d). We concluded that the TME of KP tumors inhibits the intrinsic capacity of cDC1 to engulf dying cells. cDC1 preferentially cross-present tumor associated antigens acquired via phagocytosis to activate tumor specific CD8[+] T cells[5,6,24]. To understand the impact of defective uptake on the capacity to cross-present antigens, we generated a KP-OVA line expressing low levels of the model antigen OVA (OVA, clone G11, Supplementary Fig. 3A). KP-OVA tumors grew less than the parental KP tumors and showed smaller nodules at late time points (Supplementary Fig. 3b). Depletion of CD8[+] T cells during the first 2 weeks upon tumor challenge led to a dramatic increase in tumor burden indicating that tumor containment was mediated by CD8[+] T cells (Supplementary Fig. 3c). Staining with the 25D-1.16 antibody to detect cross-presented peptide:MHC OVA on tissue cDC1 showed high antigen density in early KP-OVA tumor bearing lungs and low signal in late tumor bearing lungs (Fig. 1e and Supplementary Fig. 3d). To further probe changes in the capacity to cross-present tumor associated antigens we isolated cDC1, cDC2, and AM from mice bearing early or late KP-OVA tumors. In early tumors, cDC1 were the only subset capable of inducing efficient activation of OVA specific CD8, in line with previous reports (Fig. 1f and Supplementary Fig. 3e)[3,5,6,36]. This property was significantly inhibited in cDC1 from late tumors indicating that cross-presentation of antigens picked-up in tumor tissue is lost in advanced tumors (Fig. 1f). To confirm suppression of cross-presentation of cell-associated antigens, we delivered OVA-loaded apoptotic thymocytes in the lung. CD8 T cells proliferated in control and early tumor bearing animals but not in in late tumor bearing animals (Fig. 1g). Spontaneous IL-12 production in vivo was slightly decreased yet cells responded efficiently to ex-vivo restimulation suggesting that cytokine signals by DC1 are not deeply compromised (Supplementary Fig. 3f, g). In addition, peptide pulsed cDC1 induced robust activation of OVA specific CD8[+] T cells, indicating normal interaction with T cells and no impairment in co-stimulatory signals (Supplementary Fig. 3i). Cross-presentation of soluble protein antigens delivered ex-vivo was diminished in late tumor cDC1, in line with previous reports[13] (Supplementary Fig. 3j). Thus, by dissecting individual steps of DC1 activity, we evidenced a primary defect in antigen capture by tumor tissues resident cDC1.

**Lung cDC1 fail to activate CD8[+] tumor responses in late tumors.** To examine whether reduced antigen presentation by cDC1 along with tumor progression correlates to impairment of

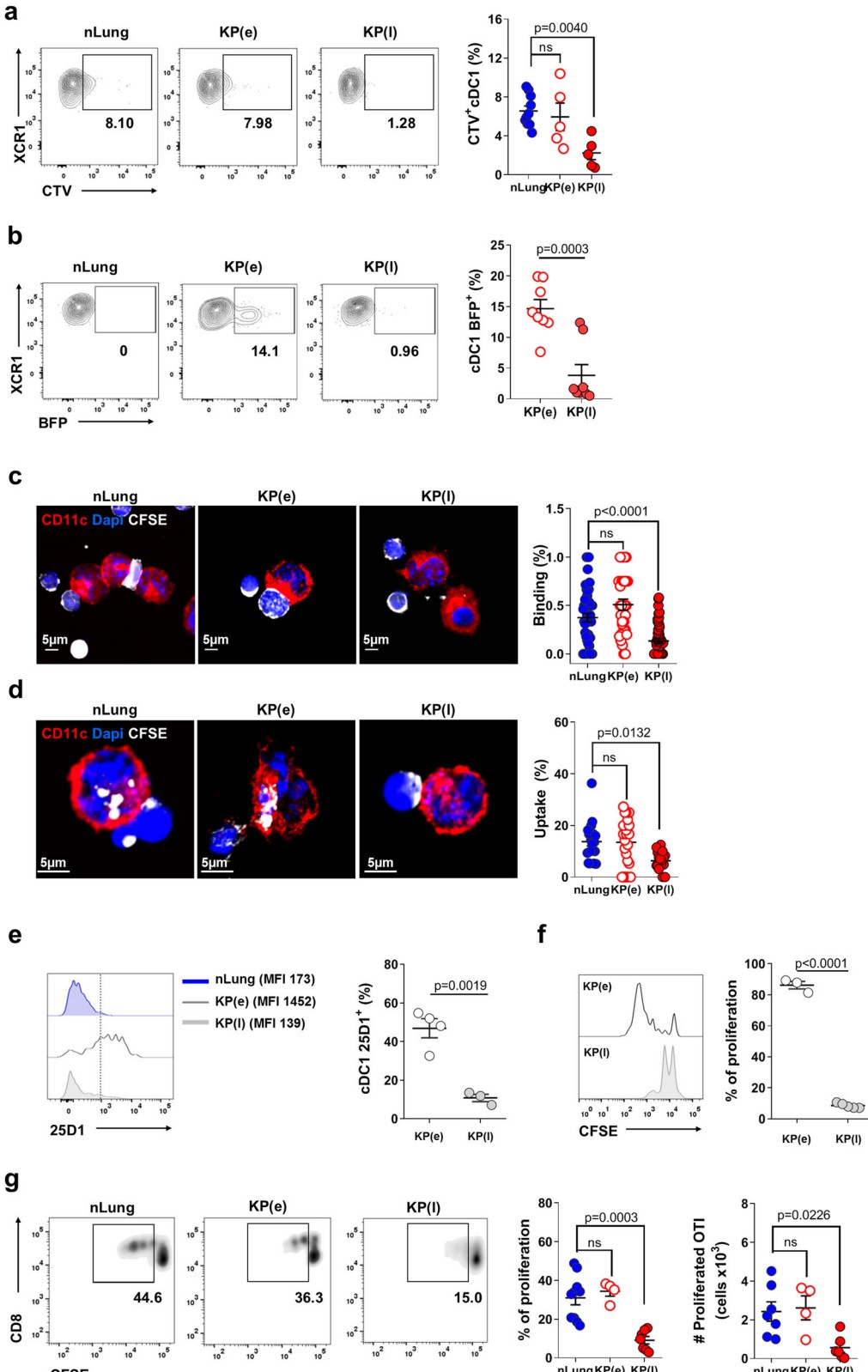

CD8+ T cell responses in vivo, we tracked endogenous CD8+ T cell over time in the blood of animals challenged with KP-OVA. Circulating IFN-γ+ CD8+ T cells were observed by day 14 after challenge, peaked at day 21 and declined to almost baseline by day 28 (Fig. 2a, b). The same trend was observed in lung tissue CD8+ T cells when comparing early and late time points (Fig. 2c). Immunohistochemistry showed high density of CD8+ T cells in

early nodules as opposed to few CD8+ T cells confined at the tumor border in late tumors (Fig. 2d). Because loss of T cell responses may have resulted from selection of OVA negative tumor cells in late tumors[33], we verified OVA expression by flow cytometry and immunohistochemistry. The percentage of OVA expressing CD45− cells was similar in early and late tumors (Supplementary Fig. 4a). The intensity of OVA staining was

**Fig. 1 Uptake of cell associated tumor antigens by cDC1 is inhibited in advanced lung tumors. a** Control mice (nLung), or mice carrying early (e) or late (l) KP tumors were challenged i.t. with CTV labeled apoptotic thymocytes. Two hours later total lung cell suspensions were analyzed by flow-cytometry to assess the fraction of phagocytes associated to fluorescence. Representative flow cytometry plots of cDC1 (gated on $CD45^+$/$Cd11c^+$/SiglechF$^-$/MHC-II$^{high}$/CD11b$^-$) and quantification of uptake, expressed as percentage of CTV$^+$ cDC1. Data were examined over two independent experiments: nLung $n = 8$, KP(e) $n = 5$, KP(l) $n = 5$, one-way ANOVA followed by Tukey's post-test. **b** Mice were challenged with KP-BFP cells. Lungs were harvested at early or late time points after challenge to prepare total lung cell suspensions. Representative flow cytometry plots displaying the percentage of BFP$^+$ cDC1 and the corresponding quantification. Data are pooled from two experiments with $n = 8$ mice per group, two-tailed unpaired $t$-test. **c, d** Sorted-cDC1 cells from nLung, early or late whole tumor tissues were incubated for 30′ (binding, **c**) or 2 h (uptake, **d**) with CFSE labeled apoptotic thymocytes. **c** Representative micrographs showing cDC1 binding to thymocytes. The percentage of cDC1 in contact with at least one thymocyte was quantified on 30 individual cells acquired on separated fields in two independent experiments. **d** Representative micrographs showing thymocytes uptake. The percentage of cDC1 that has engulfed at least one cellular fragment is plotted as uptake. Data are representative of 30 cells/condition in two independent experiments. In **c** and **d** one-way ANOVA followed by Tukey's post-test. **e** Tumors were induced by i.v. injection of KP-OVA expressing cells. Cross-presentation by lung cDC1 was detected by flow cytometry on whole lung tissues by labeling with peptide:MHC OVA specific antibody (25D1.16) (gated on $CD45^+$/$Cd11c^+$/Siglech-F$^-$/MHC-II$^{high}$/CD11b$^-$). Representative histograms and quantification of the percentage of positive cells based on the gate depicted on histograms. Data are from two independent experiments: KP(e) $n = 4$, KP(l) $n = 3$ (cDC1 were pooled from two mice for analysis), two-tailed unpaired $t$-test. **f** cDC1 were sorted from whole lung cell suspension of early and late KP-OVA bearing tumors and mixed with CFSE-labeled OVA specific T cells (OT-I). Representative CFSE peaks and quantification of proliferation (cells undergoing at least two cycles of proliferation over total T cells). Each point corresponds to cDC1 isolated from KP(e) $n = 3$ or KP(l) $n = 4$ mice in three independent experiments, two-tailed unpaired $t$-test. **g** CSFE labeled OT-I were transferred into nLung, early, and late tumor bearing animals. OVA-loaded apoptotic thymocytes were administered i.t. and T cell proliferation in the mediastinal lymph node (MLN) was evaluated 2 days later. Data are from two independent experiments, nLung $n = 9/7$, KP(e) $n = 4$ or KP(l) $n = 7$, one-way ANOVA followed by Tukey's post-test. All data are expressed as mean ± SEM. Source data are provided as a Source Data file.

similar in small nodules in early tumors and large nodules in late tumors (Supplementary Fig. 4b), indicating that the antigen is still expressed in advanced lesions. To distinguish cell intrinsic exhaustion of CD8$^+$ T cells from defective antigen presentation, we analyzed proliferation of naïve CFSE labeled CD8$^+$ OVA specific T cells transferred into early and late tumor bearing-lungs. Proliferation was induced efficiently in early tumors but not in late tumors, suggesting lack of priming by APC in situ (Fig. 2e, f). In addition, proliferation was reduced in Batf3$^{-/-}$ mice[3], indicating that cDC1 are required for full T cell activation in this model (Fig. 2g, h). Together these data indicate that inefficient priming of tumor specific CD8$^+$ T cells in established tumors coincides with loss of antigen presentation by tissue resident cDC1.

**Gene expression profiling shows loss of TIM4 in lung tumor associated cDC1.** To explore the molecular basis of loss of antigen acquisition in late tumor, we compared the transcriptional profile of cDC1 isolated from control lung (nLung) or late KP tumor-bearing lungs. Factors defining the cDC1 lineage were similarly expressed in control and tumor associated KP cDC1, indicating that cDC1 maintained their identity (Fig. 3a). Gene expression analysis returned more than 40 differentially expressed transcripts (false discovery rate ≤5% and absolute fold change ≥2; Supplementary Fig. 5a and Supplementary Table 1). Transcripts enriched in tumor-associated cDC1 belonged preferentially to the interferon alpha pathway, intracellular trafficking and antigen processing (Supplementary Fig. 5b–d). We focused on phagocytic receptors to find a mechanism for loss of antigen uptake. As shown in Fig. 3b, c; *Timd4*, encoding for the PtdSer receptor TIM4[25], was highly expressed in nLung cDC1 and strongly downregulated in late tumors. Other receptors implicated in binding and engulfment of apoptotic cells (*Clec9a, Cd36, Axl, Lrp11, Ly75, Stab1, MerTK, Marco, Tyro3, Scarf1*) showed no significant modulation, with the exception of *Cd300lf*, a second receptor for PtdSer known to inhibit efferocytosis in DCs[37], and *Cd209a* (DC-SIGN), which was slightly down-modulated. Since selective high expression of TIM4 by lung cDC1 had never been reported we validated our data by analysis of publicly available datasets. Tabula Muris single cells dataset from normal lung tissues (*Nature*. https://doi.org/10.1038/s41586-018-0590-4) confirmed co-expression of *Timd4* with markers of cDC1 in a cluster of myeloid cells (Fig. 3d). Importantly, analysis of a

recently generated scRNA dataset from control and KP tumor murine lungs[16] confirmed selective expression of TIM4 in the cDC1 subset and a significant decrease in tumors condition (Fig. 3e). Flow cytometry confirmed high expression of TIM4 protein in lung cDC1 and lack of expression in the other lung phagocytic subsets (Supplementary Fig. 6a, b). Further profiling across multiple mouse tissues[38] showed the expected high expression in tissue resident macrophages, in a consistent fraction of Langerhans cells in the skin and in a fraction of dermal cDC1[32]. Slight TIM4 expression was present on liver cDC1, on a small fraction of migratory cDC1/cDC2 in mediastinal LNs and in splenic and bone marrow cDC1/cDC2, confirming the highest expression on lung cDC1 in naïve mice (Supplementary Fig. 6c, d).

To verify the kinetic of TIM4 downregulation in tumors we analyzed early stages that correspond to functional cDC1. We observed a slight, not significant, trend of reduction (Fig. 3f), providing a causal link between receptor levels and engulfment capacity (Fig. 1a, b). Interestingly, orthotopic KP injection in one lobe was sufficient to induce a slight reduction in the contra-lateral lobe (Supplementary Fig. 7a). Expression was not detected in any of the other lung phagocytes in late tumors, including TAMs, nor in the stroma or tumor cells (CD45 negative fraction) (Fig. 3g), in contrast to previous data in breast tumors and melanoma[29,31]. A slight reduction in TIM4 was observed also in early autochthonous KP tumors (Supplementary Fig. 7b), and in cDC1 associated to Lewis lung carcinoma tumors (Supplementary Fig. 7c).

**TIM4 on lung cDC1 control uptake of cell-associated antigens.** TIM4 plays a non-redundant role for tethering and engulfment of apoptotic cells in peritoneal macrophages[27,39]. Whether TIM4 may function as a receptor for dead-cell uptake in the DCs compartment or in cDC1, is poorly documented[28]. To analyze the role of TIM4 during engulfment of BFP-KP cells, we first verified exposure of phosphatidylserine by tumor cells growing in lung tissues (Supplementary Fig. 7d). We next administered TIM4 blocking antibodies or isotype controls during the initial phases of tumor engraftment. Receptor blockade was effective and did not alter maturation or absolute numbers of phagocytes excluding bystander effects (Supplementary Fig. 7e). TIM4 blockade caused a significant reduction in the fraction of cDC1 engulfing BFP tumor cells, without affecting the uptake by the other subsets (Fig. 4a and Supplementary Fig. 7f). We further tested the impact

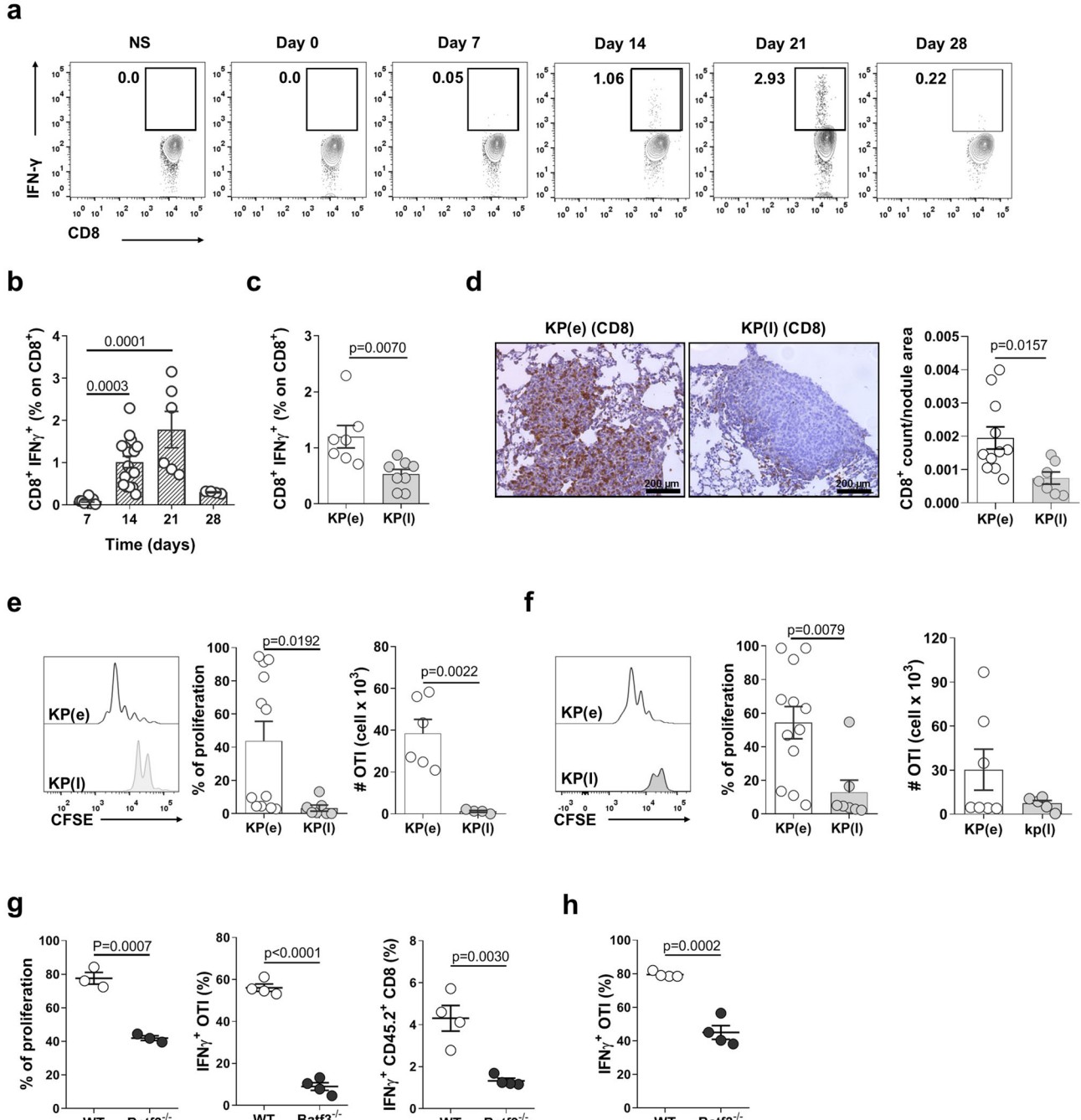

**Fig. 2 CD8$^+$ T cell responses and cross-presentation of tumor antigens by cDC1 are inhibited in late tumors. a, b** Representative plots and quantification of the kinetic of IFN-γ production by endogenous blood CD8$^+$ T cells in KP-OVA challenged mice. Data are from two independent experiments at day 7–28 or 3 independent experiments at day 14–21. In **b** Kruskal–Wallis followed by Dunn's post-test day 7 $n = 6$, day 14 $n = 15$, day 21 $n = 6$, day 28 $n = 4$ mice. **c** Percentage of IFN-γ producing endogenous CD8$^+$ T cells in lungs bearing early or late KP-OVA tumors. KP(e) $n = 7$, KP(I) $n = 8$ mice in two independent experiments, two-tailed unpaired $t$-test. **d** CD8$^+$ T cells localization in early and late KP-OVA tumors. Scale bar 200 μm. Quantification of the density of CD8$^+$ T cells as a function of the nodule area was calculated automatically on three consecutive sections/sample. Data are from KP(e) $n = 11$, KP(I) $n = 7$, two-tailed unpaired $t$-test. **e**, **f** Proliferation profile of OT-I in the MLN (**e**) or lung (**f**) of mice bearing early or late KP-OVA tumors was measured 2 days after transfer. Proliferation index (cells that underwent at least two cycles of proliferation) and absolute numbers of proliferated cells are depicted. In **e** % of proliferation KP(e) $n = 12$, KP(I) $n = 7$; absolute numbers KP(e) $n = 6$, KP(I) $n = 4$. In **f** %proliferation KP(e) $n = 12$, KP(I) $n = 7$; absolute numbers KP(e) $n = 7$, KP(I) $n = 5$. Data are from three independent experiments, two-tailed unpaired $t$-test. **g**, **h** KP-OVA tumors were induced in WT and Batf3$^{-/-}$ mice. **g** Proliferation and IFN-γ production of adoptively transferred OVA specific CD8 T cells and IFN-γ production by endogenous CD8$^+$ T cells in the lung. % proliferation WT $n = 3$, Batf3$^{-/-}$ $n = 3$; IFNγ$^+$OTI (%) and IFNγ$^+$CD45.2$^+$CD8(%) WT $n = 4$, Batf3$^{-/-}$ $n = 4$; two-tailed unpaired $t$-test. **h** IFN-γ production by OT-I in MLNs. Data represent one of two independent experiments ($n = 4$ mice/group), two-tailed unpaired $t$-test. All data are expressed as mean ± SEM. Source data are provided as a Source Data file.

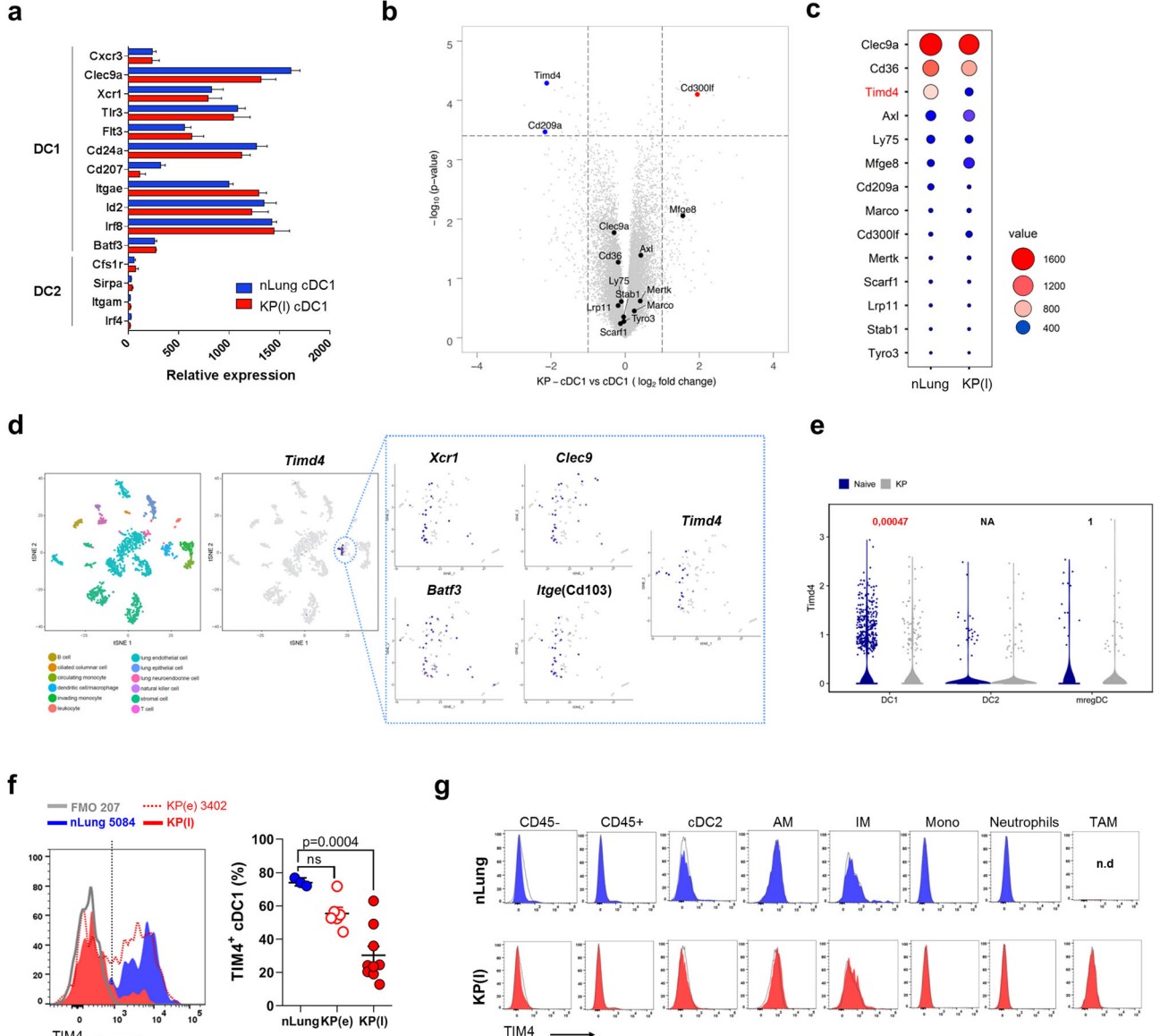

**Fig. 3 Selective cDC1 expression of the PtdSer receptor TIM4 in lung phagocytes and downregulation in tumors. a** cDC1 were isolated from nLung or late KP tumors to analyze transcriptional profiles. The graph shows expression of cDC1 and cDC2 lineage specific genes in nLung-sorted and tumor-sorted cDC1. Data are expressed as mean ± SD with nLung n = 4, KP(l) n = 3. **b** Volcano plot showing the fold change of genes encoding scavenger receptors (significantly modulated genes are depicted in red and blue, using a q-value < 5%, a false discovery rate (FDR) ≤ 5% and a fold change ≥2 or ≤2), using the Significance Analysis of Microarray (SAM) algorithm. **c** The dot-plot shows the relative abundance of genes in **b** using a color and size scale of linear expression values. **d** T-distributed stochastic neighbor embedding (tSNE) plots of single cells expression profiles in mouse lungs (Tabula Muris). Colors represent clusters of different subsets based on global similarity of gene expression. The cluster corresponding to DC-macrophages was further analyzed to assess co-expression of cDC1 markers and *Timd4* transcripts, as depicted in the boxes. **e** scRNA seq of naive and KP lung tumors from[16] were analyzed to visualize *Timd4*-expressing cells in three subpopulation of lung DCs. Violin plots show expression of *Timd4* in cDC1, cDC2, and mregDCs. Statistical analysis was performed using *Seurat* R package, taking in consideration both number of expressing cells and levels of expression. **f** Representative histograms and quantification of the percentage of cells expressing TIM4 in nLung, early and late lung tumors (the gate for identification of positive and negative cells is depicted in the histogram). Data are from nLung n = 3, KP(e) n = 6, KP(l) n = 9 mice. Significance was determined by one-way ANOVA followed by Tukey's post-test; data are expressed as mean ± SEM. **g** Representative histograms of TIM4 expression on total CD45⁻ and CD45⁺ cells and all lung phagocytic subsets in nLung and late KP lungs. Alveolar macrophages (AM), interstitial macrophages (IM), tumor associated macrophages (TAM), monocytes (Mono). Each peak is overlaid on the corresponding FMO controls (black line). Source data are provided as a Source Data file.

of TIM4 blockade on phagocytosis of dying thymocytes. TIM4 blocking antibodies in nLung selectively reduced the uptake by cDC1 almost to the level found in KP tumors, and had no additive effect when administered to tumor lungs (Fig. 4b) (Supplementary Fig. 7g). Uptake was reduced as well in cDC1 isolated from wt or Timd4⁻/⁻ mice and incubated with apoptotic thymocytes labeled with pH Rodo, a probe that becomes

fluorescent only under acidic conditions[39,40] (Fig. 4c). Blockade or genetic deletion of TIM4 impacted similarly on the subsequent step of antigen presentation. Blocking antibodies decreased antigen presentation of KP-OVA antigens (peptide:MHC OVA complexes labeling) by tissue cDC1 (Fig. 4d), and proliferation of adoptively transferred CD8 T cells into mice receiving OVA loaded apoptotic thymocytes (Fig. 4e). Finally, cDC1 isolated

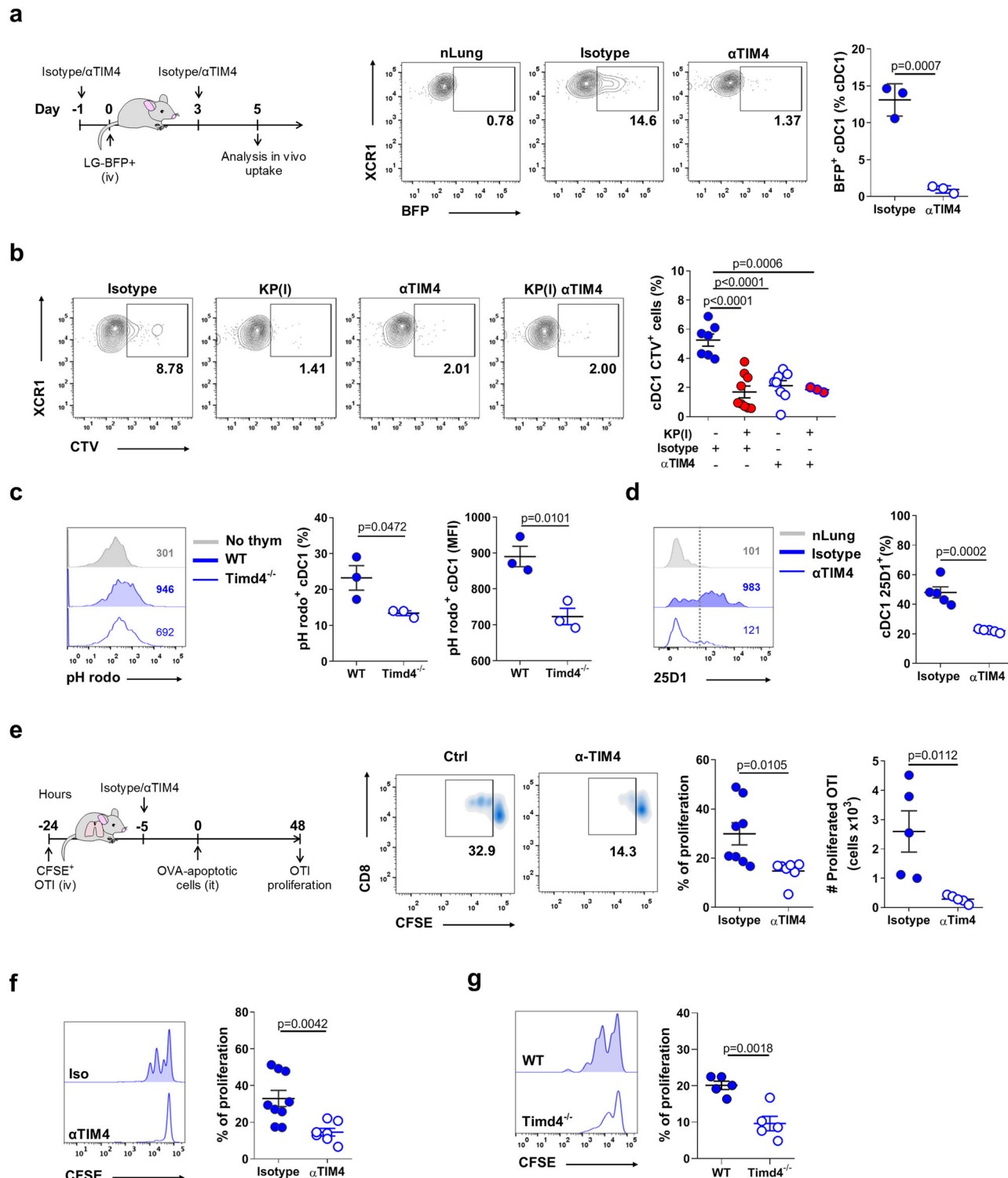

from early KP-OVA lungs of animals treated with TIM4 blocking Abs failed to induce proliferation of OVA specific T cells ex-vivo (Fig. 4f). The same reduction was observed when using Tim4$^{-/-}$ animals as recipient (Fig. 4g). We concluded that TIM4 expression in lung cDC1 is required to capture and present cell-associated antigens in normal lungs and early tumor tissues.

**TIM4 controls initiation of anti-tumor CD8 immunity.** To directly assess the impact of TIM4-mediated engulfment on

anti-tumor CD8 responses we administered TIM4 blocking antibodies during tumor challenge. TIM4 blockade prevented induction of OVA specific endogenous effector CD8$^+$ T cells in the peripheral blood of KP challenged mice (Fig. 5a), and reduced the activation of adoptively transferred OVA specific T cells (Supplementary Fig. 7h). To further investigate the impact of TIM4 during T cell priming we inoculated KP-OVA tumors into wild-type or Timd4$^{-/-}$ animals. Numbers of IFN-γ$^+$ CD8$^+$ T cells at day 7 after challenge were significantly decreased in Timd4$^{-/-}$ mice as compared to WT controls (Fig. 5b). Moreover, absolute numbers of total CD8$^+$ T cells

**Fig. 4 TIM4 controls capture and cross-presentation of cell-associated antigens in lung resident cDC1. a** Mice were challenged with KP-BFP and treated with blocking antibodies to TIM4 or isotype control as depicted. The uptake by lung cDC1 was quantified 5 days after tumor challenge. Dot plot and quantification of one representative of two experiment with $n = 3$ mice/group are shown. **b** Control mice or late KP tumor bearing mice were pre-treated with isotype control or αTIM4 antibodies and challenged with CTV-labeled apoptotic thymocytes. The percentage of lung phagocytes associated to fluorescence was evaluated by flow-cytometry 2 h after challenge. Representative plots and quantification of two independent experiments are shown, Isotype $n = 7$, Isotype KP(l) $n = 9$, αTIM4 $n = 8$, αTIM4-KP(l) $n = 3$. **c** Uptake of pH rodo loaded thymocytes by lung cDC1 in WT and Timd4$^{-/-}$ mice. Representative histograms and quantifications (percentage and MFI of pH rodo$^+$ cDC1) for one of three independent experiments with $n = 3$ mice/group are shown. **d** Cross-presentation by lung cDC1 (labeling with 25D1.16 antibody specific for MHC class-I:OVA peptide complex) in day 7 KP-OVA bearing animals treated with TIM4 blockade. Representative histograms showing MFI values and quantification of five animals/group in two independent experiments. **e** In vivo proliferation of adoptively transferred OT-I induced by i.t. injection of OVA-loaded apoptotic thymocytes, in the presence of TIM4 blocking antibodies or isotype control as depicted in the scheme. Graphs show frequencies and absolute numbers of proliferated CD8$^+$ T cells in the MLN (at least one cycle of proliferation). Data are representative of two independent experiments. % proliferation Isotype $n = 8$, αTIM4 $n = 7$; Absolute numbers five mice/group. **f** Ex-vivo OT-I proliferation induced by cell-sorted cDC1 from day 7 KP-OVA lung tumors under TIM4 blockade or isotype treatment. Representative dilution profile and quantification from three independent experiments (each point corresponds to the pool cDC1 from three mice), Isotype $n = 9$, αTIM4 $n = 7$. **g** As in **f**, cDC1 were isolated from day 7 KP-OVA lung tumors induced in WT or Timd4$^{-/-}$ animals ($n = 5$ mice in two independent experiments). All data are depicted as mean ± SEM. Significance was determined by one-way ANOVA followed by Tukey's post-test for **b**, or two-tailed unpaired $t$-test. Source data are provided as a Source Data file.

and OVA pentamer CD8$^+$ were diminished in the lung of Timd4$^{-/-}$ mice (Fig. 5c, d). To overcome potential bystander effects of systemic TIM4 blockade and pinpoint precisely on cDC1, we generated mixed bone marrow chimeras using mixtures of wild-type:Batf3$^{-/-}$ or Timd4$^{-/-}$Batf3$^{-/-}$ bone marrow (Supplementary Fig. 8a–c). IFN-γ production and accumulation of CD8$^+$ T cells was decreased in animals carrying a specific deletion of TIM4 in the cDC1 compartment (Fig. 5e, f). Most importantly, containment of tumor growth was reduced in Timd4$^{-/-}$:Batf3$^{-/-}$ chimeras as shown by more abundant lesions and larger nodules than in animals reconstituted with wild-type:Batf3$^{-/-}$ bone marrows (Fig. 5g). Thus, selective ablation of TIM4 in cDC1 reduces protective anti-tumor responses. We finally tested whether TIM4 contributes to rejection induced by immunogenic chemotherapy, which favor APC activation[41,42]. We used a combination of oxaliplatin and cyclophosphamide (oxa/cycl), previously shown to induce CD8$^+$ T cell responses and rejection in this same KP model[34]. Animals were challenged with KP tumors and treated with oxa/cycl at day 7 and 14 after challenge (Fig. 6a). The drug increased CD8/CD4 ratio both in LNs and lung tissues and significantly reduced tumor burden with respect to untreated animals (Fig. 6b–d). The efficacy of oxa/cycl treatment was abrogated in Batf3$^{-/-}$ mice (Fig. 6e), indicating that responses induced by immunogenic therapy require cDC1 in this model. Of note, addition of TIM4 blocking antibodies one day before oxa/cycl administration resulted in reduced efficacy of immunogenic therapy (Fig. 6b–d), denoting that TIM4 plays a role during capture of drug-induced apoptotic cells to induce anti-tumor responses.

**TIM4 expression in lung adenocarcinomas correlates with signatures of protective immune cells and predicts responses to PD-1 checkpoint blockade.** To investigate the relevance of our observations in human cancers, we explored TIM4 expression in the Cancer Genoma Atlas (TCGA) dataset. We found a correlation between Timd4 transcripts and a gene signature specific for human cDC1 and CD8 effector T cells in early stage patients (Fig. 7a) (Supplementary Table 2 and refs. [7,8]). To further explore the potential prognostic impact of TIM4, we analyzed Kaplan–Meier survival curves in the TCGA dataset of lung adenocarcinoma. Interestingly, early-stage patients expressing a high cDC1 signature complemented by Timd4 showed a small yet significant advantage in survival (Fig. 7b). Based on these observations, we reasoned that TIM4 may be important to determine responsiveness to therapies that promote CD8 responses. To this goal, we analyzed a recent dataset comparing the transcriptional profiles before treatment of responders and non-responders to anti PD-1 immunotherapy (GSE126044). Remarkably, Timd4

expression was significantly higher in patients who responded to therapy than in non-responders (Fig. 7c), suggesting that TIM4 confers an advantage under the selective pressure of immunotherapy. A cDC1 signature and a CD8 effector signature had a similar predictive power of positive responses to therapy (Fig. 7d), indicating that Timd4 could be an additional biomarker to identify tumors with intact antigen presentation.

## Discussion
Cells of the myeloid lineage have been delineated with increasing precisions in the TME revealing their complexity, plasticity, and key roles in promoting or restricting tumor growth. Among these, cDC1 have emerged as invariably linked to positive outcomes across multiple preclinical models and human tumors[5–8,16]. A key function enabling cDC1 anti tumoral properties is the efficient capture and internalization of tumor cell fragments for cross-presentation to CD8$^+$ T cells[5,6,19–21,36,43,44]. Here, we show that this exquisite cDC1 property is targeted by the microenvironment of lung adenocarcinoma. Past investigations to explain suppression of DCs activity in the TME have mostly focused on post internalization events like processing of soluble antigen, cellular metabolism, reduced maturation[10–12,14,45]. In contrast, the mechanism underlying acquisition of tumor antigens in tissues, and its modulation during tumor progression have been poorly explored. By comparing cDC1 resident in healthy lung tissues to those exposed to early or late lung tumors, we have been able to trace the evolution of antigen capture as a function of tumor progression. We uncover that cDC1 are key to sample tumor antigens and initiate tumor specific CD8 responses in early lesions. Instead, at later tumor stages, antigen acquisition become inefficient, coinciding with loss of antigen presentation and fading of T cell responses. Thus, a major hurdle to sustain tumor specific CD8 responses is suppression of antigen acquisition by the most efficient cross-presenting subset in tissues.

By analyzing expression of scavenger receptors, we identified the loss of the PtdSer receptor TIM4 as one major mechanism underlying cDC1 deactivation in lung murine tumors. Additional cell intrinsic changes, such as defective cross-presentation and impaired cytokine production, likely contribute to overall cDC1 suppression. However, by impinging on the initial step of antigen acquisition, we propose that loss of TIM4 may dominate over downstream events. The uptake of dead cells has been attributed to selective expression of receptors, that are capable to bind to and promote engulfment of dying cells in cDC1[19–24]. These studies were mostly performed using lymphoid resident cDC1 and model antigens, whereas the receptors involved in

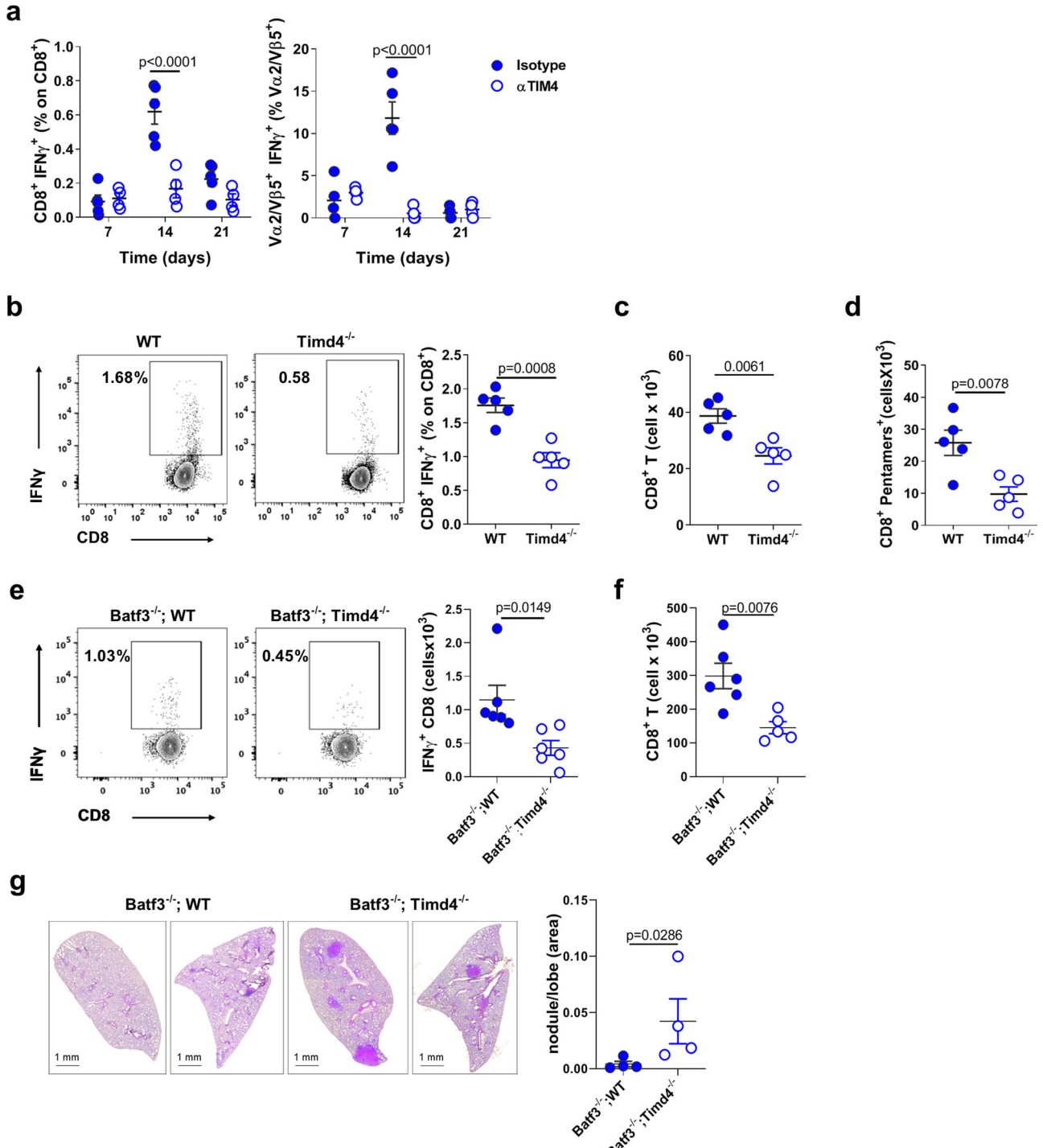

**Fig. 5 TIM4 promotes anti-tumor CD8+ T cell responses and tumor control. a** Effect of TIM4 blockade on activation of endogenous CD8+ T cells in the blood of KP-OVA challenged mice. The percentage of IFN-γ+ CD8 and OVA specific-IFN-γ+ CD8 (Vα2/Vβ5+) as a function of total CD8 and Vα2/Vβ5+ T cells, respectively, at various time points after challenge. Isotype $n = 5$, αTIM4 $n = 4$, data represent one of two independent experiments. **b–d** Activation of anti-tumor T cell responses in WT and $Timd4^{-/-}$ animals 7 days after challenge with KP-OVA. **b** Representative plots and frequencies of IFN-γ+ endogenous CD8+ T cells in MLN. **c** Absolute numbers of CD8+ T cells and **d** OVA-specific CD8+ T cells (pentamers+) accumulated at tumor site. Data are representative of two experiments ($n = 5$ mice/group). **e–g** Lethally irradiated CD45.1 WT mice were injected with a 1:1 mixture of CD45.2 bone marrows of the indicated genotypes to generate bone marrow chimeras. Ten weeks later mice were inoculated with KP-OVA tumors. **e** Representative plots and frequencies of IFN-γ+ CD8+ T cells in MLN 10 days after tumor challenge ($n = 6$ mice/group). **f** Absolute numbers and quantification of CD8+ T cells accumulated at tumor site with Batf3$^{-/-}$;WT $n = 6$ and Batf3$^{-/-}$;Timd4$^{-/-}$ $n = 5$ mice. **g** Tumor growth in bone marrow chimeras was assessed 28 days after tumor challenge. Representative sections and quantification of tumor nodules calculated automatically on four consecutive sections of three lobes/animal and expressed as a function of the total lobe area. Each value in the plot represents the average of the three lobes and four sections for each mouse. All data are mean ± SEM. Significance was determined by two-way ANOVA followed by Sidak's post-test for **a**, two-tailed unpaired *t*-test for **b–f**, and two-tailed Mann–Whitney test for **g**. Source data are provided as a Source Data file.

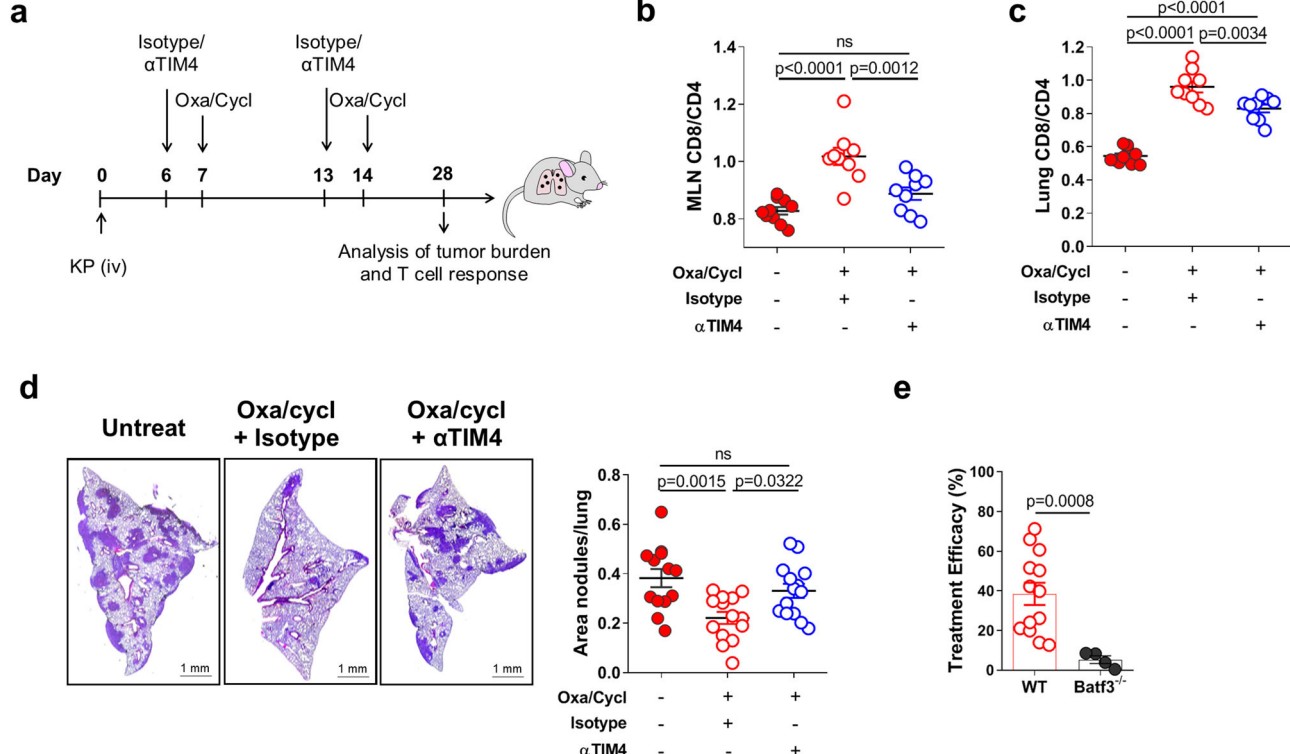

**Fig. 6 TIM4 is required for tumor rejection induced by immunogenic chemotherapy. a** Scheme of experimental design. Mice were challenged with KP cells and left untreated or treated with oxa/cycl and anti-TIM4 or isotype as depicted in the scheme. **b, c** Analysis of the CD8/CD4 T cell ratio in the MLNs (**b**), Untreated $n = 10$, Oxa/Cycl+Isotype $n = 9$, Oxa/Cycl+αTIM4 $n = 9$; and in the lungs (**c**), Untreated $n = 10$, Oxa/Cycl+Isotype $n = 10$, Oxa/Cycl+αTIM4 $n = 9$; of mice 2 weeks after the last treatment. Data are from two independent experiments. **d** Effect of TIM4 blockade on the efficacy of the immunogenic chemotherapy. Tumor burden was assessed at day 28 by calculating the area covered by tumor nodules on four consecutive sections of three lobes/animal and expressed as a function of the total lobe area. Results are from untreated $n = 13$, Isotype+Oxa/Cycl $n = 14$ and Oxa/Cycl+αTIM4 $n = 14$ in two independent experiments. **e** WT and Batf3$^{-/-}$ mice were challenged with KP cells and treated with oxa/cycl or left untreated. Treatment efficacy is expressed as the percentage of reduction in tumor area over not treated control calculated in 14 WT and 4 Batf3$^{-/-}$ mice. All data are expressed as mean ± SEM. Significance was determined by one-way ANOVA followed by Tukey's post-test for **b**–**d** and two-tailed Mann–Whitney test in **e**. Source data are provided as a Source Data file.

acquisition of tumor antigens by cDC1 resident in different tumor tissue have been less investigated. TIM4 binding to PtdSer in macrophages is key to engulf and clear dying cells and its genetic deletion promotes development of autoimmunity[40,46–48]. The pattern of expression and specific function of TIM4 in the DCs compartment has remained controversial, likely because of a lack of proper definition of DCs subset in past analysis[26,28,29,49–51]. To fill this gap, we performed an extensive characterization of DCs subsets across mouse tissues using recently established gating strategies[38]. Besides the well-established expression in most tissue resident macrophages, we found a selective expression of TIM4 in mouse lung cDC1 that was higher than in any other tissues DCs. Curiously, alveolar lung macrophages are among the few resident macrophages that do not express TIM4. It will be extremely intriguing to identify the tissue factors that control transcriptional and epigenetic regulation of TIM4 expression[52,53] in steady state myeloid cells. A second important issue relates to the function of TIM4 in tumors and the mechanism behind receptor downregulation. TIM4 expression was reported to be upregulated in macrophages and DCs infiltrating melanoma and mammary mouse tumors, and to be associated with suppression of immune responses in colorectal mouse models[29,31,54]. In contrast, mouse lung tumors do not induce recruitment of myeloid cells expressing TIM4 but rather suppress its expression on cDC1, ablating T cell priming. These observations suggest a highly specific tissue and context

dependent regulation, that would be important to investigate in more tumor types and human tumors. A recent report showed decrease of TIM4 and consequent loss of clearance in aging macrophages caused by p38 activation[55], suggesting a possible mechanism of receptor loss in cDC1.

In this study, we used blocking antibodies and genetic deletion to demonstrate the key role of TIM4 in engulfment of dying cells and tumor cells and cross-presentation of cell associated antigens by lung cDC1. Moreover, TIM4 blockade inhibited responses to a combination of immunogenic drugs previously shown to induce T cell responses in this same tumor model[34,56], indicating its role in promoting processing of released antigens. Together these evidences underscore the role of TIM4 in cDC1 as a PtdSer receptor implicated in capture of tumor antigens in cDC1, which contributes to surveillance of nascent tumors and initiation of tumor protective responses in mouse lung tumors. These findings somehow conflict with a previous study showing that TIM4 induces degradation of tumor antigens by autophagy, thereby inhibiting immunity[29]. This discrepancy likely arises from the different antigen processing machinery in bone-fide cDC1 as opposed to macrophages or CD11c/MHC-II$^{high}$ cells and the different tumor tissue analyzed.

Gene expression from the Cancer Genoma Atlas (TCGA) showed a correlation between Timd4 transcripts and a gene signature for human cDC1[8], and effector CD8 T cells and better survival for patients expressing high levels of cDC1 and Timd4. In

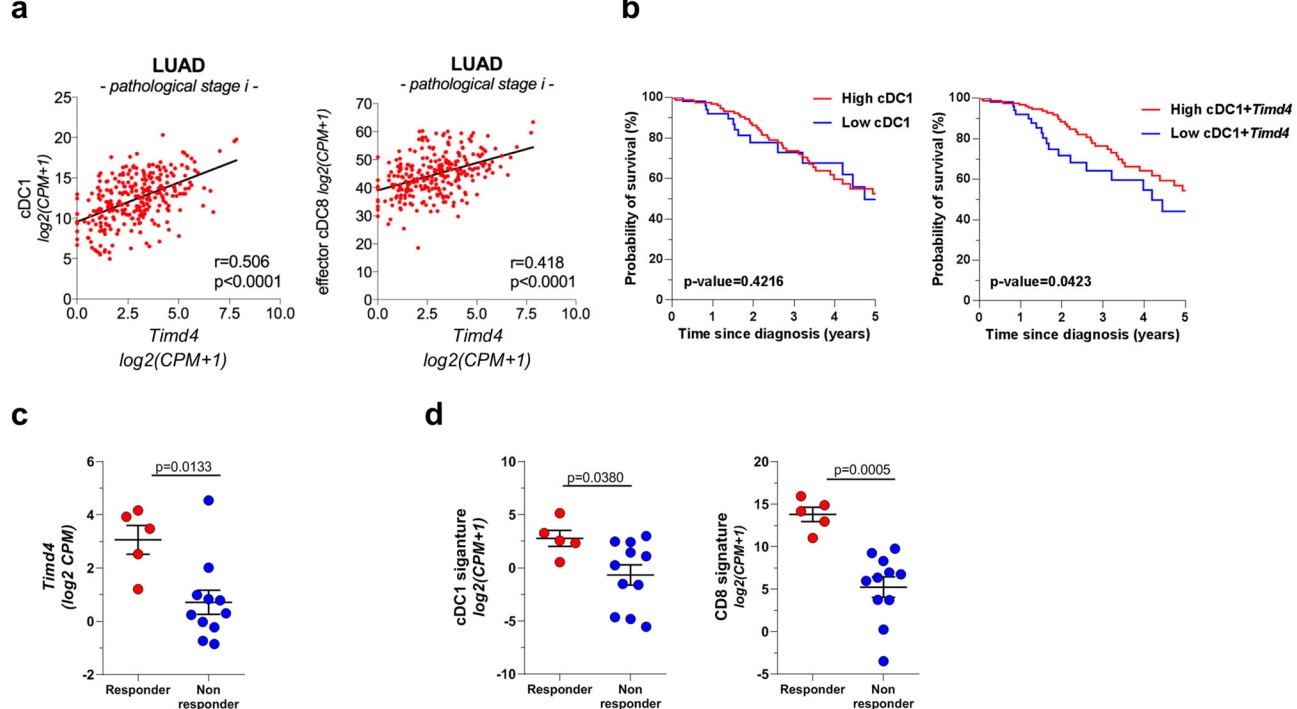

**Fig. 7 Timd4 expression correlations and predictive value in human lung adenocarcinoma. a** Correlation between *Timd4* transcripts and gene signatures for cDC1 and cytotoxic T cells (gene signatures described in Supplementary Table 2) within lung adenocarcinoma cases (LUAD) in TCGA. Pearson correlation coefficient in the *cor.test* function of R *stats* package with *two.sided* as alternative hypothesis. **b** Kaplan–Meier survival curves showing probability of survival for stage-I LUAD cases from the TCGA dataset (n = 219). The two groups with either high or low expression of cDC1 or cDC1 + Timd4, were identified based on the classification rule described in the ref. [60] and detailed in methods. Survival curves were compared using the log-rank (Mantel–Cox) test. **c**, **d** Patients from cohort GSE126044 that responded or not to PD-1 therapy were classified based on baseline pre-treatment expression of *Timd4* (**c**), cDC1 and CD8 cytotoxic T cells signatures (**d**). P values are calculated using the two-tailed Mann–Whitney test; responder n = 5, Non-responder n = 11. Source data are provided as a Source Data file.

addition, a small trial showed a better response to checkpoint blockade in patients, suggesting an association between TIM4 expression and protective immune responses in human lung tumors. Further analysis of scRNA-seq data, flow cytometry and immunohistochemistry are needed to establish the precise pattern of expression across human tissues and its significance for the human disease.

In conclusion, this study identifies TIM4 as a lung cDC1 phagocytic receptor implicated in immune surveillance of early murine tumors, that is targeted at later stages of tumor progression impairing antitumor responses. Initial evidences support the potential of TIM4 as a biomarker for stratification of human lung tumors.

## Methods

**Mice**. C57BL/6 mice were purchased from Envigo Laboratories and maintained in sterile isolators at the ICGEB animal Bioexperimentation facility. Ethical and experimental procedures were reviewed and approved by the ICGEB Animal Welfare board, meeting the requirements of the EU Directive 2010/63/EU. Timd4$^{-/-}$ mice (Miyanishi et al.[40]) were maintained under pathogen-free conditions at Osaka University and Experiments were approved by the Animal Care and Use Committee of the Research Institute of Microbial Diseases, Osaka University and part of the colony was transferred at ICGEB. The following strains were from Jackson Laboratories: Kras$^{G12D}$ p53$^{-/-}$ (DuPage et al.[33]); B6.129S(C)-Batf3$^{tm1Kmm}$/JBatf3$^{-/-}$; OT-I and OT-II (C57BL/6-Tg (TcraTcrb). These strains were maintained under pathogen-free conditions at Brescia University, King' College, and in sterile isolators at ICGEB, respectively. OTI were crossed to congenic CD45.1 mice for tracking experiments. Experiments on the Kras$^{G12D}$ p53$^{-/-}$ strain were approved by the Italian Ministry of Health (approval number 165/2017-PR, issued on 20/02/2017). All experiments performed at the ICGEB animal facility were approved by Italian Ministry of Health (approval number 1155/206-PR and 646/2019-PR).

**Cell lines**. The transplantable KP tumor model has been isolated from primary lung tumors of C57BL/6 KP mice (K-ras$^{LSLG12D/+}$; p53$^{fl/fl}$ mice)[57]. The line called (LG1233) was kindly provided by Dr. Tyler Jacks (Massachusetts Institute of Technology, Cambridge, USA) in July 2015. Generation of the cell line is described in the ref. [57]. The cell line was not authenticated. To generate KP-OVA cells, KP cells were transduced with the lentiviral vector Pdual-liOVAha-PuroR, encoding Ii-OVA (kindly provided by David Murrugarren, CIB, Navarra). After puromycin selection, cells were subcloned and single cell clones were tested for HA expression by intracellular staining using anti-HA antibody (clone 3F10, Roche). To generate KP-BFP cells the KP line has been transduced with a pAIP-BFP lentiviral vector. This vector was obtained by adding XbaI and NotI sites to the BFP gene from plasmid pU6-(BbsI) CBh-Cas9-T2A-BFP-P2A-Ad4E1B (Addgene plasmid 64218), and inserting it into the lentiviral vector pAIP (Addgene plasmid 74171) under the control of the SFFV promoter. The Lewis lung carcinoma cell line 3LL (LL2, LLC/1) has been purchased by the ATCC (CRL-1642) in October 2014.

All cell lines were maintained in DMEM media supplemented with 10% fetal bovine serum (FBS, Euroclone) and Gentamicin (50 μg/mL, Gibco) and routinely tested for mycoplasma contamination. Cells were expanded to passage 3 and stored in aliquots in liquid nitrogen. Cells used for in vivo challenge were been passed less than five passages.

**Establishment of orthotopic lung tumors**. In order to establish lung orthotopic tumors, 1 × 10$^5$ KP, 5 × 10$^5$ KP-OVA, or 5 × 10$^5$ KP-BFP cells were injected intravenously in 100 μL PBS.

For KP and KP-BFP tumors early and late time points were defined as day 10–14 and 28–35. This definition of tumor stages is applied to all experiments with the exception of some analysis, that were anticipated to day 5 or 7 as specified in legends.

To establish 3LL tumors, 1 × 10$^4$ 3LL cells were mixed at a 1:1 ratio with Geltre LDEV-Free Reduced Growth Factor Basement Membrane Matrix (Gibco) in a final volume of 20 μL, and injected into the left lung lobe by intercostal injection at the median axillary line of C57BL/6 mice. For local delivery of KP cells, 1 × 10$^4$ KP cells were injected as above and tissues were collected at day 28 after engraftment. As ctrl, a group of mice was injected with PBS:Matrigel without tumor cells.

**Induction of autochthonous lung tumors**. KP mice (K-ras[G12D/+];p53[fl/fl] mice) at 8 weeks of age, were intranasally inoculated with $2.5 \times 10^7$ infectious particles of a replication-deficient adenoviral vector with Cytomegalovirus promoter driving the expression of the Cre recombinase protein in order to sporadically induce mutations in lung cells, and promote lung tumor development as previously described (DuPage et al.[33]). Mice were sacrificed 10 weeks after the adenovirus inoculation, lungs were collected upon intracardiac perfusion with cold PBS and mechanically and enzymatically treated as already described.

**Lung tumor dissociation and dendritic cell isolation**. Lung tissue, control or infiltrated by tumors, were harvested after PBS lung circulatory perfusion. The whole tissue was mechanically cut into small pieces and digested with Collagenase type 2 (265 U/mL; Worthington) and DNase (250 U/mL; Thermo scientific) at 37 °C for 30'. Collagenase was then stopped by EDTA 10 mM and the cell suspension was filtered using 70 μm cell strainer (Corning).

CD11c[+] lung cells were separated by immunomagnetic sorting using CD11c microbeads (Miltenyi Biotec) following manufacturer's instructions. Dendritic cells were purified by FACS sorting with FACS Aria II (BD Biosciences), using the gating strategy shown in Supplementary Fig. 1c, after CD11c[+] cells staining in PBS + 2% FBS with the following antibodies: CD11c-A647 (N418), CD103-PE (2E7) purchased from Biolegend and SiglecF-BB515 (E50-2440), CD11b-BV421 (M1/70) and MHCII-BV711 (M5/114.15.2) from BD Biosciences. For gene expression analysis up to 30,000 cells were sorted directly into 350 μL RLT buffer (Qiagen). For ex-vivo stimulation, 10,000 cDC1 were plated in U-bottom 96-well, stimulated with CpGB (1 μg/mL, Invivogen) or DMXAA (25 μg/mL, Invivogen) for 3 h and then lysed in RLT buffer for gene expression analysis.

**Liver and skin dissociation**. Livers were harvested after perfusion with 10 mL of cold PBS, mechanically cut into small pieces and digested with collagenase IV (0.2 mg/mL) and DNase I (0.03 mg/mL) 37 °C for 30 min. Collagenase was then stopped by adding 3 mL of PBS + EDTA 10 mM and the cell suspension was filtered using 70 μm cell strainer (Corning). Red blood cells were lysed by using RBC lysis buffer (Biolegend).

Mice ears were harvested and split into dorsal and ventral halves to expose the inner dermal layer. Both halves were laid dermid side down in dispase solution (Dispase 4 U/mL) 1 h at 37 °C. After incubation, the tissue was cut into small pieces and digested 1 h at 37 °C with collagenase IV (0.2 mg/mL) and DNase (0.03 mg/mL).

Cell suspensions were then processed for FACS staining.

**Flow cytometry**. The antibodies used for the experiments are listed in Supplementary Table 4 and panels used to identify subsets are illustrated in Supplementary Table 5. Viability was assessed by staining with LIVE/DEAD Fixable Aqua Dead Cell Stain Kit (Life Technologies) or Annexin V/7AAD (Biolegend). For cell staining, FcR binding sites were blocked by using αCD16/CD32 (93, Biolegend). Samples were then stained with specific antibodies in PBS + 1% BSA and fixed with PBS + 1% PFA. To verify in vivo production of IL12p40, ctrl or KP-bearing mice were treated with Brefeldin A (Sigma) iv for 6 h and then sacrificed. Lungs were harvested, dissociated, and stained for the identification of lung-resident cDCs. During all process cells have been maintained in the presence of brefeldin A. For intracellular staining of OVA-HA, KP-OVA cells or lung cell suspensions were fixed and permeabilized using Cytofix/Cytoperm solution (BD Biosciences) following manufacturer's instructions, and then stained with rat monoclonal anti-HA (3F10, Roche) and then with anti-rat A647 (Thermo Fisher).

Absolute cell count was performed by adding TrueCount Beads (Biolegend) to the samples following manufacturer's instructions, or by multiplying the percentage of the identified cell type by the total number of cells in each tissue. The total number of cells were obtained by counting viable cell numbers with Trypan blue (Gibco). After dead cell exclusion and exclusion of cell doublets, the following populations were identified according to the gating strategy provided in Supplementary Fig. 1c, d. For annexin labeling of tumor cells, annexin staining was added to single cell suspension of lungs bearing day 7 KP-BFP tumors according to manufacturer's instructions and immediately analyzed.

Flow data were acquired with FACS Celesta (BD Biosciences) and analyzed with Diva software (BD Bioscience) or FlowJo software (Tree Star, Inc.).

**TIM4 blockade, CD8 depletion, and immunogenic chemotherapy**. For TIM4 blocking studies healthy or tumor-bearing mice were treated with anti-TIM4 antibody (RMT-53, BioXCell) or isotype control (2A3, BioXCell) at the concentration of 250 μg/100 μL intravenously or intraperitoneally. The protocol for TIM4 blockade (number of injections and timing) are described in each specific experiment.

For CD8 depletion C57BL/6 mice were injected with anti-mouse CD8 (YTS 169.4, BioXCell) or isotype control (2A3, BioXCell) at the concentration of 200 μg/100 μL intraperitoneally at day −1, +4, 8, 11 after KP-OVA implantation. Mice were sacrificed 15 days after tumor inoculation to assess tumor burden and validate CD8[+] T cell depletion by FACS.

KP tumor-bearing mice were either left untreated or received chemotherapy intraperitoneally (i.p.) at day 7 and 14 after tumor engraftment (Oxaliplatino,

2.5 mg/kg; Cyclophosphamide, 50 mg/kg, Tocris Bioscience). In these experiments, mice were treated with anti-TIM4 antibodies or isotype control the day before chemotherapy administration.

**Detection of specific OVA-MHC-I complexes on cDC1**. To detect specific MHC class-I OVA complexes CD11c[+] cells were isolated from KP-OVA early or late tumors by microbeads and incubated for 1 h at 37 °C with PE-coupled 25-D1.16 mAb (BioLegend). Cells were washed and stained with extracellular antibodies to identify cDC1, cDC2, and AM and analyzed by flow cytometry. As negative control, CD11c[+] cells were isolated from control lungs. For TIM4 blocking studies, mice were treated at day −1 and 3 post KP-OVA inoculation with TIM4 blocking antibodies or isotype control and lungs were harvested at day 7 for labeling as above.

**Uptake of KP-BFP cells**. Mice were challenged with KP-BFP tumor cells and lung tissues were collected at different time points as indicated in the legends. Total lung cell suspensions were stained to identify phagocytes, and the fraction of each subset associated to BFP fluorescence was used to calculate the uptake index. To compare uptake at early and late time points data were normalized against the numbers of BFP cells in the lung tissue. In TIM4 blocking experiments, mice were treated with anti-TIM4 antibodies or isotype control at day −1 and 3 w KP-BFP challenge and lungs were harvested at day 5 post-tumor inoculation. Total lung cell suspensions were labeled to identify BFP[+] phagocytes. The frequency of phagocytes associated to BFP fluorescence was assessed by flow cytometry and normalized on the frequency of total CD45[−] BFP[+] cells.

**In vivo uptake, ex-vivo binding and engulfment of dying thymocytes**. Thymocytes were isolated from Balb/C animals and incubated for 5 h at 37 °C in RPMI + 10% FBS medium with Dexamethason (5 μM, Tocris). After incubation the percentage of apoptotic cells assessed by AnnexinV-BV421/7AAD staining was higher than 60%. For in vivo uptake, apoptotic cells were stained with 5 μM of CellTrace™ Violet Cell Proliferation Kit (CTV, Thermo Fisher Scientific). After staining, cells were resuspended in PBS at the concentration of $20 \times 10^6$ in 20 μL, and injected intratracheally in control mice or mice bearing early or late KP tumors. In TIM4 blocking experiments, C57BL/6 mice were treated the day before apoptotic cell administration with anti-TIM4 antibody or isotype control (ip). Lungs were harvested 2 h after apoptotic cell administration, dissociated and stained for the identification of CTV[+] lung resident phagocytes.

For in vitro analysis of internalization, apoptotic thymocytes were labeled with 5 μM of CFSE (BioLegend), washed extensively and co-cultured for 2 h with cells sorted cDC1 derived from nLung, early or late KP tumor bearing lungs at 3DC:1thymocytes ratio. Cells were plated on fibronectin coated coverslips for 1 h. For in vitro binding, isolated cDC1 derived from nLung or KP tumor bearing lungs were first adhered to fibronectin-coated coverslips and co-cultured for 30 min with thymocytes at 1DC:10thymocytes ratio. Uptake and binding were analyzed by confocal microscopy using a 60× planar objective on a Zeiss 880 instrument. To calculate the phagocytic index the fraction of cells containing at least one fragment of green material within the intracellular space was scored, and divided by the total number of DC on the coverslip. DCs were distinguished as they label positive for CD11c used for cell sorting. To calculate binding, the fraction of DC showing at least one contact with a green T cells was counted and divided by the total number of DC on the coverslip.

**Uptake of dying cells by Timd4[−/−] lung phagocytes**. After perfusion with HBSS/2 mM EDTA, lungs were processed using a GentleMACS dissociator (Miltenyi Biotech) according to the manufacturer's instruction, and treated at 37 °C for 30 min with 2 mg/mL Collagenase D (Roche Diagnostics) and 100 μg/mL DNase I (Worthington) in DMEM containing 10% FBS. After passing through a 70-μm cell strainer (BD Biosciences), cells were collected by centrifugation at $300 \times g$ for 5 min, and suspended in PBS containing 2% FBS and 2 mM EDTA. Dead and unwanted cells were removed using EasySep™ Biotin Selection Cocktail (STEMCELL), and biotinylated mouse MFG-E8 D89E, and the following biotinylated antibodies: CD31 (390) and TER-119 (TER-119) from BioLegend, and CD3e (145-2C11), CD19 (1D3) and Ly-6G/Ly-6C (RB6-8C5) from BD Biosciences. The remaining cells were plated on 24-well plates at $2 \times 10^5$ cells per well. The $2 \times 10^6$ pHrodo-labeled apoptotic thymocytes that had been prepared as described (Yanagihashi et al.[39]) were added to the well, and incubated at 37 °C for 2 h. After washing with PBS, the cells were stained with the antibodies described below, suspended in 20 mM CHES-NaOH buffer (pH 9.0) containing 150 mM NaCl and 2% FBS, and analyzed by flow cytometry. The antibodies used are anti-CD45-APC/Fire750 (30-F11, BioLegend), anti-SiglecF-PerCP/eFluor710 (1RNM44N, Thermo Fisher Scientific), anti-CD11c-APC (N418, BioLegend), anti-MHC II (I-A/I-E)-Alexa-Fluor488 (M5/114.15.2, BioLegend), anti-CD103-PE/Cy7 (2E7, BioLegend), and anti-XCR1-BV510 (ZET, BioLegend).

**Ex vivo priming of OT-I cell by cell sorted cDC1**. For ex-vivo T cell activation assay, cDC1 isolated from early or late KP-OVA tumors were plated in U-bottom 96 wells and co-incubated with CFSE-labeled OTI T cells at the ratio 1:5 ($10^4$ cDCs and $5 \times 10^4$ CD8[+]OTI). After 48 h of co-culture OTI T cell proliferation was assessed by flow cytometry. For TIM4 blocking experiments, mice were treated the

day before and 3 days after KP-OVA inoculation with anti-TIM4 antibodies or isotype ctrl. In these experiments, mice were sacrificed 7 days post-tumor engraftment and isolated lung-resident cDC1 or total mediastinal lymph node cells were incubated with CD8+OTI cells. For T cell priming assay using cDC1 pulsed ex-vivo DC1 isolated from nLung or late KP tumor-bearing lungs were plated $1 \times 10^4$ in U-bottom 96-well, stimulated with polyI:C (1 μg/mL) and pulsed with class I OVA peptide or OVA protein at the indicated doses. After 3 h stimulation, cells were washed twice in PBS and co-incubated with $1 \times 10^5$ OT-I T cells. IL-2 production was detected in the supernatant after 16 h of co-incubation by ELISA Max Standard sets (Biolegend), following manufacturer's instructions.

**Adoptive transfer and activation of OTI.** $1 \times 10^6$ KP-OVA cells were injected iv in wt or Batf3$^{-/-}$ mice to establish tumors. At day 14 post challenge $1 \times 10^5$ CD45.1+OTI CD8+ OVA-specific T cells were labeled with CFSE (5 μM, Biolegend) or CTV (5 μM, Thermo Fisher Scientific), and intravenously injected into mice bearing early or late tumors. After 2 days from the injection mice were sacrificed and lungs and MLN were collected, processed as described previously and proliferation was verified by flow cytometry. In Batf3$^{-/-}$ experiments total lung cells were re-stimulated ex-vivo with OVA class-I peptide SIINFEKL (2 μM), and stained intracellularly with anti-IFN-γ antibody using the intracellular staining kit BD Golgi STOP. In TIM4 blocking experiments, mice inoculated with KP-OVA cells were treated at day −1 and +3 post-tumor inoculation with anti TIM4 or isotype control, injected with CTV-labeled OTI cells at day 4 and sacrificed at day 7 to analyze OTI accumulation and proliferation in MLN.

**Cross-presentation of OVA loaded tymocytes.** CD8+ T cells were isolated from lymph nodes of OTI mice by beads separation (CD8a+ T cell isolation kit, Miltenyi Biotec) and labeled with CFSE (5 μM, Biolegend) following manufacturer's instructions. $1 \times 10^6$ CFSE-labeled CD8+ OTI cells were injected i.v. in healthy or early and late KP-bearing mice. The day after OTI injection, Balb/C thymocytes were isolated and incubated for 1 h at 37 °C (RPMI + 10%FBS + β-mercapthoethanol) with OVA protein (1 mg/mL) at the concentration of $10^7$/mL. After 1 h, Dexamethason (5μM) was added to the cell suspension and cells were incubated 4–5 h at 37 °C. Apoptosis was evaluated by AnnexinV/7AAD staining. Apoptotic cells (apoptosis > 90%) were injected intra-trachea ($20 \times 10^6$ in 20 μL) with PolyI:C (1 μg/mouse). In TIM4 blocking experiments, mice were iv injected with anti-TIM4 antibody or isotype control 5 h before apoptotic cells administration (250 μg/mouse). Cross-presentation of cell associated antigens was assessed 48 h after intratracheal injection, by verifying OTI accumulation and proliferation in mediastinal lymph node by flow cytometry.

**Activation of endogenous anti-tumor CD8+ T responses.** For the detection of endogenous anti-tumor CD8+ T cells in KP-OVA bearing mice, blood was collected at day 7, 14, 21, and 28 post-tumor inoculation. After red blood cell lyses, cells were stimulated with SIINFEKL (2 μM) 37 °C for 4 h in the presence of Golgi Stop (BD Biosciences) to allow accumulation of intracellular cytokines. After viability and surface marker staining, cells were fixed and permeabilized using Cytofix/Cytoperm solution (BD Biosciences) following manufacturer's instructions, and then stained with αIFNγ-PE (XMG1.2, Biolegend). For the analysis of the activation state of endogenous tumor-infiltrating CD8+ T cells, KP-OVA tumor bearing lungs were harvested at early and late time points, dissociated and cell suspensions were processed as described above for evaluation of IFNγ production.

Timd4$^{-/-}$ mice were injected with $2 \times 10^5$ KP-OVA cells and sacrificed 7 days post-tumor inoculation. We analyzed IFNγ production in lung-draining lymph nodes and the accumulation of endogenous anti-tumor CD8+ T cells in the lungs by using R-PE labeled Pro5® MHC H-2KbPentamers (Proimmune) following manufacturer's instructions. Briefly, total lung cell suspension was stained with R-PE labeled Pro5® MHC H-2KbPentamers for 45 min at 4 °C, washed and stained to identify CD8 using the gating strategy shown in Supplementary Fig. 1d.

Endogenous T cell response was also analyzed as described above in BM chimeric mice. BM chimeras were injected with KP-OVA cells and sacrificed at 10 days post-tumor inoculation. Lung draining LNs were ex vivo stimulated and stained for IFNγ production, and lungs were processed and stained for assessing the endogenous CD8+ T cell accumulation at tumor site.

**Bone-marrow mixed chimeras.** To generate bone marrow mixed chimeras, CD45.1 mice were lethally irradiated with 450 rad followed by a second dose of 450 rad 4 h apart. The day after irradiation mice were reconstituted with $5 \times 10^6$ total BM cells of a 1:1 mixture (Timd4$^{-/-}$: Batf3−/− or wt: Batf3$^{-/-}$) by intravenous injection. Before we initiated the experiments, mice were allowed to engraft for 10 weeks and engraftment was assessed by analyzing the percentage of CD45.2 (donor cells) over the CD45.1 (recipient cells) in the blood. Animals were then injected iv with KP-OVA cells ($1 \times 10^6$) and sacrificed at early (10 days) or late (28 days) time points for analysis. To check the chimerism, lung-resident cDC1 and peritoneal macrophages were stained for TIM4 expression.

**Real time PCR.** Total RNA was extracted from isolated cDC1 by using the RNeasy MicroKit (Qiagen) according to manufacturer's instruction. cDNA was synthesized using SuperScript VILO (Invitrogen) and real-time PCR for gene expression was

performed using SsoFast EvaGreen Supermix (Biorad) with specific primers for IL12a, IL12b, Ifna, and Ifnb are listed in Supplementary Table 3.

**Immunohistochemistry on mouse samples.** To evaluate tumor burden lung tissues were harvested, fixed in formaldehyde 10% and paraffin embedded following standard procedure. Consecutive sections every 200 μm were dewaxed and rehydrated and stained with the H&E (Bio-Optica, Milano Spa). The area of tumor nodules was quantified over consecutive sections and averaged (three sections/sample). Measurements and automatic thresholding were performed using Image J. The area occupied by tumor nodules was expressed as a function of the total lung lobe area. For CD8+ T cells infiltration and HA expression, sections were treated with antigen-retrieval solution (Vector laboratories) for 20 min at 120 °C. Slides were treated for 2 min in $H_2O_2$ to block endogenous alkaline phosphatase. After blocking in 10% goat serum in 0.1% Tween20 for 30 min slides were incubated overnight in a humidified chamber in a 1:500 dilution of anti-mouse CD8 (Invitrogen cat 14–0808–82) in blocking buffer or anti-HA 1:50 (Roche). Detection was performed using the ImmPRESS polymer detection system (Vector Laboratories), according to manufacturer's Instructions. Ilastik software was used to automatic detect and segmentate CD8+ cells. Segmented images were further automatic analyzed for the number of foci using Image J software. The number of tumor-infiltrating CD8+ T cells was normalized on tumor nodule area.

**Gene expression profiling of ex vivo purified cDC1 cells.** Gene expression profiling was performed on Mouse Gene 2.0 ST arrays (Affymetrix, Santa Clara, CA, USA). Total RNA was extracted using TriPure (Roche) from ex vivo purified cDC1 cells from the lungs of healthy and KP tumor-bearing mice ($n = 4$ and $n = 3$ independent samples, respectively). RNA quality and purity were assessed on the Agilent Bioanalyzer 2100 (Agilent Technologies, Waldbronn, Germany); RNA concentration was determined using the NanoDrop ND-1000 Spectrophotometer (NanoDrop Technologies Inc.). RNA was then treated with DNaseI (Ambion). In vitro transcription, hybridization and biotin labeling were performed according to Affymetrix 3'IVT protocol (Affymetrix).

All microarray data analyses were performed in R (version 3.4.2) using Bioconductor libraries (BioC 3.5) and R statistical packages. Probe level signals were converted to expression values using robust multiarray average procedure RMA of Bioconductor *affy* package, and a custom definition file for mouse Gene 2.0 ST arrays based on Entrez genes from BrainArray (version 22.0.0; http://brainarray.mbni.med.umich.edu/Brainarray/Database/CustomCDF/22.0.0/entrezg.asp). Raw data are available at Gene Expression Omnibus under accession number GSE119574.

To identify differentially expressed genes, we compared the expression levels of ex vivo purified cDC1 cells from the lungs of healthy and KP-tumor bearing mice using the Significance Analysis of Microarray (SAM; algorithm coded in the *samr* R package[58]. In SAM, we estimated the percentage of false positive predictions (i.e., false discovery rate, FDR) with 100 permutations and selected those probe sets with FDR ≤ 5% and absolute fold change larger than a selected threshold (e.g., ≥2) in the comparison of cDC1 cells from tumor-bearing and healthy mice (Supplementary Table 1).

Global unsupervised and supervised clusterings were performed using the function *hclust* of R *stats* package with Pearson correlation as distance metric and average agglomeration method. Gene expression heatmaps have been generated using the function *heatmap.2* of R *gplots* package after row-wise standardization of the expression values. The volcano plot, showing the most significantly differentially expressed genes in the comparison of ex- vivo purified cDC1 cells from tumor and healthy mice, was generated using the *ggplot* function of the *ggplot2* R package. *P*-values were derived from SAM *q*-values using the function *samr.pvalues.from.perms* of the *samr* R package (Supplementary Table 1). Functional over-representation was performed using Gene Set Enrichment Analysis (GSEA; http://software.broadinstitute.org/gsea/index.jsp) and the curated gene sets of the Molecular Signatures Database (MSigDB) derived from the Hallmark pathway collection (http://software.broadinstitute.org/gsea/msigdb/collections.jsp). Prior to GSEA analysis, we converted mouse Entrez IDs into the corresponding human homologous genes using the HUGO Gene Nomenclature Committee (HGNC) database (https://www.genenames.org/cgi-bin/hcop). Gene sets were considered significantly enriched at FDR ≤ 0.25 when using Signal2Noise as metric and 1000 permutations of gene sets. The dot plot, showing the most significantly enriched and depleted gene sets in cDC1 sorted from lungs with established KP tumors, was generated using the *ggplot* function of the *ggplot2* R package. Immgen data were downloaded from the Consortium webpage and linear expression values for Timd4 were replotted in Graphpad.

**Collections of lung adenocarcinoma.** Gene expression data (RSEM gene normalized) and clinical information of the TCGA lung adenocarcinoma (LUAD) were downloaded from the Broad GDAC Firehose Portal and used as is. Transcriptional profiles of non-small-cell lung cancer patients responding and non-responding to anti-PD-1 treatment were downloaded from Gene Expression Omnibus GSE126044[59]. Raw counts have been converted to CPM and log-CPM values using the *cpm* function of the *edgeR* package in R.

Average signature expression has been calculated as the standardized average expression of all signature genes in sample subgroups (e.g., histological grade, response to therapy). The gene sets used for the signatures and the relevant literature are illustrated in Supplementary Table 2. To identify two groups of tumors with either high or low expression of a gene set, we used the classification rule described in the ref. [60]. Briefly, we classified tumors as "low" if the gene set score was lower than the 25th percentile of the distribution of the gene set score and as "high" otherwise. This classification was applied to the expression values of the TCGA LUAD dataset, early stages. To evaluate the prognostic value of a gene set, we estimated the overall survival probability using the Kaplan–Meier method and compared the Kaplan–Meier curves using the log-rank (Mantel–Cox) test. $P$ values were calculated according to the standard normal asymptotic distribution. Survival analysis was performed in GraphPad Prism.

*Single-cell RNA-seq data analysis.* Murine lung scRNA-seq data have been obtained from Tabula Muris[61]. Briefly, we downloaded from the Tabula Muris website (https://tabula-muris.ds.czbiohub.org/) the R object file containing cell annotations and clustering information of cells analyzed using FACS-based cell capture in plates. tSNE plots of all cell types and of the "dendritic cell/macrophage" subset have been generated using *Seurat* R package[62]. 3D scatterplots were generated with the *plot3d* function of the *rgl* R package. Control and KP tumor murine lungs scRNA-seq data were obtained from the ref. [16]. Briefly, we downloaded from the GitHub repository associated to the publication (https://github.com/effiken/Maier_et_al_nature_2020) the R object file containing expression values and cell annotations. Differential expression analysis was performed using *Seurat* R package.

**Statistic.** Primary data were collected in Microsoft Excel and statistical analysis were performed using Graphpad Prism 6. For comparison between two or more groups with normally distributed datasets two-tailed Student's $T$-test, one-way ANOVA or two-way ANOVA were used as appropriate. The non-parametric Kruskal–Wallis test with Dunn's multiple comparison was performed to compare three or more unmatched groups. $P$ values ≤ 0.05 were considered significant.

**Reporting summary.** Further information on research design is available in the Nature Research Reporting Summary linked to this article.

## Data availability

All microarray raw data generated for the present study, along with gene expression matrices and metadata for each sample, are publicly available in Gene Expression Omnibus (GEO) under accession code [https://www.ncbi.nlm.nih.gov/geo/query/acc.cgi?acc=GSE119574]. RSEM gene normalized data and clinical information of the TCGA lung adenocarcinoma (LUAD) were downloaded from the Broad GDAC Firehose Portal [http://firebrowse.org/?cohort=LUAD&download_dialog=true]. RNA-seq raw counts of non-small-cell lung cancer patients responding and non-responding to anti-PD-1 treatment were downloaded from GEO [https://www.ncbi.nlm.nih.gov/geo/query/acc.cgi?acc=GSE126044]. The scRNA-seq data of murine lung were obtained from Tabula Muris [https://tabula-muris.ds.czbiohub.org/] as R object file containing cell annotations and clustering information of cells analyzed using FACS-based cell capture in plates. The scRNA-seq data of control and KP tumor murine lungs were downloaded from https://github.com/effiken/Maier_et_al_nature_2020 as an R object file containing expression values and cell annotations [https://www.ncbi.nlm.nih.gov/geo/query/acc.cgi?acc=GSE131957]. The gene sets of the Hallmark pathway collection were downloaded from the Molecular Signatures Database (MSigDB) of the Broad Institute [http://software.broadinstitute.org/gsea/msigdb/collections.jsp; version 6.2]. Source data are provided with this paper. All other data supporting the findings of this study are available within the Article, Supplementary Information or from the corresponding authors on reasonable request.

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

## Acknowledgements

This work was supported by AIRC IG (14414 and 21636) to F.B. N.C. was supported by AIRC and ICGEB fellowship. F.S. is supported by an ICGEB fellowship. G.M.P. is supported by AIRC grant (21636) to F.B. Y.Y. and S.N. were supported by Grant-in-Aid for Scientific Research (S) for the Promotion of Science 15H05785 from the Japan Society for the Promotion of Science (JSPS) (to S.N.). S.B. was supported by AIRC Special Program Molecular Clinical Oncology "5 per mille" (Grant no. 10016 and grant n. 22759) and Italian Epigenomics Flagship Project (Epigen). O.R. is supported by Fondazione Umberto Veronesi (Post-Doctoral Fellowship 2020). We thank Debora Brescia supported by Fondazione Beretta (Brescia, Italy) for technical support. We thank Roberto Amadio and Luciano Gaston-Morosi for help with data analysis and statistics.

## Author contributions

N.C., G.M.P., and F.S. performed experiments. F.B and N.C. designed experiments. F.B. wrote the manuscript. S.B., O.R., E.M.C.M., and L.C. analyzed gene expression data. N.C., G.M.P., F.S., S.V., Y.Y., M.B., A.D.P., and T.S. performed additional experiments. R.D., C.A.D., and F.G. provided help for flow cytometry characterization of Tim4 expressing cells. S.Z., R.O., P.G., W.V., & S.N. contributed reagents and animal models.

## Competing interests

The authors declare no competing interests.
