## [Peer Review File · Nature Communications]

REVIEWER COMMENTS

Reviewer #1 (Remarks to the Author): with expertise in Tim4

In the manuscript, Caronni et al. show that cDC1 is a type of phagocytes engulfing tumor associated antigens, which is required for CD8+ T cell activation and proliferation preventing tumor progression in the KP lung tumor model. Especially, they show that Tim4, a PS receptor, exclusively is expressed in cDC1 and mediates engulfment of tumor associated antigen. In tumor progression, the level of Tim4 in cDC1 is downregulated, which causes less efficient engulfment of tumor associated fragments, resulting in less cross-presentation of tumor antigens by cDC1 to CD8+ T cells and thus decreased CD8+ T cell activation. Treatment of KP mice with an anti-Tim4 antibody or Tim-4 depleted mice show less engulfment of tumor cells by cDC1 and decreased cross-presentation and CD8+ T cell activation by cDC1.

Overall, the expression of Tim4 in cDC1 and its effects on engulfment of tumor cells by cDC1 are solid and clearly show that Tim4-mediated engulfment of tumor cells by cDC1 is related to CD8+ T cell activation. There are however several issues that need to be addressed and the molecular mechanism by which Tim4 is downregulated in tumor progression needs to be established.

Major comments.

The significance of the study is the finding that Tim4 expressed in cDC1 mediates engulfment of lung tumor cells by cDC1, which is required for cross-presentation of tumor antigens by cDC1 to activate CD8+ T cells and that the level of Tim4 is downregulated in KP late. Although the lower level of Tim4 in KP late than in KP early is clear, the molecular mechanism by which Tim4 is downregulated in KP late is not addressed. Considering the novelty and findings of the study, how Tim4 is downregulated in KP late is an important question to be addressed.

Regarding the comment above, it is unclear whether Tim4 downregulation in KP is causative for the tumor progression. In other words, it is unclear whether the tumor progression results in Tim4 downregulation or Tim4 downregulation causes the tumor progression. The authors may use Tim4 KO mice to monitor the tumor progression.

Lung tumor cells seem to be mostly ingested by alveolar macrophages and engulfment of lung tumor cells by cDC1 is much inferior to that by the macrophages (based on the authors' data). Although cDC1 is major cells to cross-present tumor antigens, the authors have not checked the possibilities that cross-presentation by other APC, i.g. alveolar macrophages, could be affected during tumor progression. Especially, the authors show that genes involved in intracellular trafficking and antigen processing are modulated in cDC1. Thus, it is still possible that the similar modulation happens to alveolar macrophages, resulting in less efficient cross-presentation and the less number of T cell in KP late. In addition, population engulfing tumor cells by cDC1 is less than 10% (about 6% for KP early and about 2.5 % for KP late, Fig 1A). Decreased engulfment by cDC1 in KP late is clear, but the basal engulfment capacity of cDC1 is very low. Could this change make the changes?

Once again, regarding to the comment above, the authors find that many genes, especially genes related to intracellular trafficking and antigen processing, are modulated in KP late. they focused on Tim4 without rationale excluding other modulated genes. The authors need to show whether trafficking or antigen processing is unaltered in KP late or to explain why they excludes trafficking and antigen-processing genes.

Tim4 is one of the best characterized phosphatidylserine (PS) receptor and mediates engulfment of apoptotic cells exposing PS on the cell surface. The authors adequately show engulfment of apoptotic thymocytes by cDC1, which is modulated by the level of Tim4 expressed in cDC1. However, it is unclear that engulfment of lung tumor cells by cDC1 occurs in a PS-dependent manner. The authors could check whether PS is exposed on dying or dead lung tumor cells or tumor fragments and PS masking molecules such as annexin V or a MFG8 mutant recapitulate the

phenotype of Tim4 KO or the anti-Tim4 antibody treatment.

Usage of the blocking antibody against to Tim4 is a good strategy to evaluate the effects of Tim-4 on cross-presentation, T cell activation and so on. Besides this approach, because the authors have Tim4 KO mice, it could be more reliable to use Tim4 KO mice rather than the blocking antibody to validate the effects of Tim4. The blocking antibody was used in most of experiment to evaluate the effects of Tim4. It is strongly recommended to use Tim4 KO mice for validation of Tim4 effects on cross-presentation, CD8+ T cell proliferation, and experiments in Figure 6.

Minor comments

The authors claim equal loading of tumor lungs with dying cells based on Fig S1G. Because no difference of engulfment by cDC2 or AM between KP early and late could result from other factors, this is inadequate

The percentage gated in the density map for KP BFP early in Figure 1B is too high to match the percentages in the bar graph in the figure.

Figure 1D. Assay for binding of apoptotic cells to phagocytes is usually performed by blocking internalization of apoptotic cells. The authors may use cytochalasin D or 4 oC incubation and need to show more cells (the images in the figure)

The authors show the MFI in the histograms and percentages in the bar graphs. The author need to show the gating in the histograms to get the percentages in the bar graphs (1e, 1f, 2e, 2f, 4c, 4d, 4e, S5, S6).

Tim4 in the study is from mice. Thus, the correct notation is Tim4, Tim-4 or Timd4. In the text and figures, they are mixed.

Alpha(α) or beta(β) are written as a or b in the some figures. The author need to carefully check.

The usage for Tim4 KO mice, Tim4^{-/-} and Timd4^{-/-}, is mixed in the text and figures.

There is a typo, TIME in page 3. TME seems to be correct instead of TIME.

Reviewer #2 (Remarks to the Author): with expertise in cDC1s and tumor immunology

In this manuscript, Caronni et al. identify a mechanism by which antigen presentation by conventional type 1 dendritic cells (cDC1s) is progressively inhibited during the course of tumor development in a model of Kras/p53 mutant lung adenocarcinoma. The authors provide evidence consistent with a model whereby loss of TIM-4 expression on cDC1s reduces their ability to uptake and cross-present tumor cell-associated antigens impairing, in turn, CD8+ T cell responses as tumors progress to late stages. This is an interesting finding of both fundamental and translational implications for the cancer immunology and immunotherapy field.

In terms of novelty, a previous study (Parekh et al. 2017) has already demonstrated that TIM-4 expression on cDC1s is required to take up apoptotic cells, although this study did not address this question in the context of early versus late stages of lung tumorigenesis. In contrast to the findings of the current manuscript, others have shown TIM-4 expression by myeloid cells is upregulated in tumors, and that TIM-4 reduces antigen presentation and inhibited tumor immunity

triggered by chemotherapy-induced tumor cell death (Baghdadi et al. *Immunity*, 2013). Caronni et al. propose a different role for TIM-4 expression on myeloid cells, specifically cDC1s, in uptake of tumor antigens and how its expression is modulated during tumor growth. Regarding the selectivity of expression of TIM-4 on cDC1s, others have already shown CD103⁺ cDC1s express higher TIM-4 than other myeloid cells (de Mingo Pulido et al. *Cancer Cell* 2018), so this message is not particularly novel. The comments below outline further analysis of the existing data and suggestions of new experiments that could strengthen the manuscript.

A major claim of the paper is that TIM-4 expression is progressively repressed during tumor development which is associated with a progressive decrease in the ability of cDC1 to uptake tumor cell-derived material. To substantiate this claim the authors should consider the following points:

- Show consistently ALL comparisons of normal lung vs early tumors vs late tumors rather than just comparing normal lung vs late tumors throughout manuscript (e.g. Fig 1C, Fig 1G, Fig 3A).
- Perform kinetics of TIM-4 downregulation across tumor progression (i.e. a time course of events, rather than just two arbitrary early and late timepoints).
- Analyze differences in TIM-4 expression and cross-presentation capacity of ex-vivo cDC1s isolated from early and late tumors rather than nLung vs late tumors
- Perform comparisons between early and late tumors in an independent model (i.e. 3LL model).

To substantiate the claim that TIM-4 plays a selective role in cDC1 the authors should show the effect of TIM-4 genetic ablation or Ab-mediated blockade in the acquisition of cell-associated material (uptake of apoptotic thymocytes or BFP-expressing KP cells) for ALL myeloid cell populations including monocytes and TAMs, not just cDC1s, cDC2 or alveolar macrophages. Related to this, could the authors change the representation of the uptake of material from BFP-expressing cells from the % of BFP⁺ over BFP⁺ CD45⁻ cells to the % of BFP⁺ cDC1 over total cDC1 instead, which is the more appropriate quantification.

The cross-presentation analysis of KP-OVA tumors in vivo and cDC1/CD8 co-cultures following TIM4 blockade should be repeated in Tim4d^{-/-} mice and chimeric mice with a specific TIM-4 deficiency in cDC1s to further substantiate the conclusion that TIM-4 blockade specifically limits cDC1 function rather than any other TIM-4 expressing cell that could also be targeted by the TIM-4 blocking antibodies. This would also exclude, among other possibilities, that the blocking Ab prevents uptake of tumor cell material in general. Similarly, the authors acknowledge that similarly to a general TIM-4 antibody blockade, TIM-4 deficiency in other cell subsets may account for the deficient CD8 responses in germline Tim4KO mice. The authors need to substantiate their data showing that the effect is largely on cDC1 and test if there are no other effects on non-cDC1 following Ab blockade. That similar defects are observed in Batf3^{-/-} does not prove this point as these mice have a general baseline defect in cDC1 and cross-presentation in general.

The authors should consider the following points to strengthen the characterization of TIM-4 expression across the manuscript.

- Regarding the exclusivity of TIM-4 expression on cDC1s, the authors look at most phagocytes including alveolar and interstitial macrophages but not TAMs (Figure 3E). In the profiling across multiple mouse tissues (Fig S5B,C) it is clear that macrophages expressed the highest TIM-4 across all tissue sites. Could the authors comment on this and show the expression levels of Timd4 transcripts in all populations analyzed by scRNAseq (Figure S5A) rather than only in the 'DC/macrophage' cluster?
- Given the expression of TIM-4 is not bimodal and thus defining positive and negative populations is challenging, the authors should also plot the MFI of TIM-4 (e.g. Figure 3C) rather than just the % of the parental population to facilitate comparisons between different settings (i.e. cDC1 from resting, early and late tumors). In this regard, Tim4d^{-/-} cells may be a more appropriate control for Flow Cytometry rather than FMO controls (e.g. Figure 3C), especially because typically auto-fluorescent cells show high levels of TIM4 expression (e.g. macrophages in Fig S5B).
- Likewise, it would be useful to see the expression levels of TIM-4 across multiple populations in

the tumor and systemically in chimeric mice to validate the system, especially because FigS7C seems to show TIM-4 expression is about 2 fold lower in peritoneal macrophages in the *Batf3*^{-/-}:*Tim4*^{-/-} chimera relative to the *WT*:*Tim4*^{-/-} control.

Regarding the experiments administering TIM-4 blockade before chemotherapy, it is not possible to say whether this is downstream of TIM-4 blockade on cDC1s or other cells.

- The authors could use the *Timd4*^{-/-}:*Batf3*^{-/-} bone marrow chimeras in the context of the chemotherapy experiments to better address this question, particularly given that others have shown opposite phenotype of TIM-4 in repressing tumor immunity triggered by chemotherapy (Baghdadi et al. *Immunity*, 2013).
- What is the number and phenotype of cDC1 following TIM-4 blockade and are they specifically targeted?

The mechanism of downregulation of TIM-4 in late but not early tumors is an interesting and potentially novel angle of the manuscript which is not addressed. Although identification of a mechanism is perhaps beyond the scope of this manuscript, the previous paper by Parekh et al. has shown that *Vps34* is responsible for TIM-4 downregulation in cDC1s. So *Vps34* could be a candidate to test and at least the authors should expand their discussion to integrate these previous findings.

The human evidence in this study is somewhat conflicting and necessitates further work. The authors claim TIM-4 correlates with cDC1s in human malignancies, yet the major claim of the murine experiments is that TIM-4 expression on cDC1s is in fact downregulated in advanced cancers. This correlation is higher in stage 1 patients than in all stages, but there is still a decent positive correlation between TIM-4 expression and DC1s in all stages group.

- The authors could analyze available lung cancer patients scRNAseq datasets to assess specificity of TIM-4 expression on cDC1s (e.g. Lambrechts et al. 2018 *Nat Med*).
- The authors could extend their analysis of the association of TIM-4 expression with outcomes from immunotherapy in other available datasets of other tumor types. If there is no association with response, this would be consistent with the author's claims that TIM-4 plays a specific role in lung cDC1s. Yet, given TIM-4 expression was shown to associated with cDC1 and CD8 T cell signatures it is likely that it will be associated with survival and response more broadly.
- What is the rationale for showing the association of TIM-4 with *GzmK* and not *GzmB*, *A*, perforin or any other mediator or effector molecule of cytotoxic immunity in Figure 7?
- The authors could test if TIM-4 is progressively downregulated in lung cancer as they imply by monitoring *Timd4* transcript levels across different lung cancer stages in TCGA.

Minor comments

- Clarification is required on figure legend Figure 1E: "Data are from 2 independent experiments with 4 mice/group. Each dot represents the pool of 2 animals".

- Figure 1E: The authors need to provide further evidence and controls to prove the MHC-I-SIINFEKL complex staining is specific (e.g. is the staining negative in other stained cells?)

- Figure 1F, 2E,F and 4F: Could the authors use a consistent method for assessing the proliferation of OT-I cells? What was the rationale for representing the data as a % of cells that divided at least once in some experiments or twice in others?

- Figure 4C: quantification of pHRodo fluorescence should be done using the MFI as this is not a bimodal distribution.

- Gating on cDCs in the lung based on *CD11c* and *MHCII* expression is not enough to identify cDCs, especially cDC2s. As such their increase (Fig 1 SE) in cDC2s could reflect an increase in some sort of myeloid cells (not cDC2).

- "Genes involved in intracellular trafficking and antigen processing were modulated in tumor-associated cDC1 which may explain defective cross-presentation". Which genes from those shown are involved in intracellular trafficking? Also, genes for antigen presentation are enriched in tumor cDC1s, in contrast with their finding that late tumour cDC1s fail to cross-present antigens. This point should be noted by the authors.

- Figure 7B: please include the p value or indicate which of the correlations is statistically significant.

- Figure S2 is representative of the late time point d28. What is the picture at the early time point?

- Spelling throughout manuscript: tymocytes, interferon alfa pathway, Oxaliplatin, Dexamethason and grammar: Page 19: were been passed, were anticipated to day 5 or 7.

Reviewer #3 (Remarks to the Author): with expertise in lung cancer - mouse models and immunology

GENERAL ASSESSMENT

This paper investigates the role of TIM4, a receptor known to mediate uptake of apoptotic bodies by macrophages. The data indicates that lung cDC1, already known to be important in anti-tumor adaptive immunity, use TIM4 in a non-redundant manner for uptake of lung tumor-associated antigens and cross-priming of anti-tumoral CD8 T-cells. This phenomenon appears to be lost as locoregional tumor load increases. In a translational part, it is suggested that TIM4 is a theranostic biomarker of durable response to PD1 blockade in NSCLC. As such the paper provides incremental insights on the role of TIM4 in adaptive immunity to tumor-associated antigens, without bringing fundamental change in thinking within the field of tumor immunology.

MAJOR COMMENTS:

The paper could have been stronger with more extensive use of the autochthonous tumor model to assess this underexplored aspect of tumor-induced immune dysfunction, which is the temporal dimension and the evolution of immune dysfunction as tumor load increases in a continuous fashion. However, as far as I understand, for most mechanistic experiments the authors revert to adoptive transfer of tumor cells expressing a model antigen. The conclusions drawn from the use of such "fast" transplantable models do not necessarily apply to more real-life spontaneous tumor models, e.g. when it comes to investigating immunoediting. With respect to this, the term "KP tumor lungs" as used throughout this report is confusing as it does not provide the reader with a distinction between lungs after adoptive transfer of KP cells (i.v. or intracostal), versus lungs with AdenoCre-induced tumors. See for example legend of Fig.1: what is exactly meant by "lungs bearing KP tumors"? Is the Cre-induced autochthonous tumor model only used in Fig S6B? Admittedly not all experiments can be performed in the autochthonous model (e.g. experiments involving genetic knockdown of Tim4^{-/-}), still there are models for instance that feature expression of a model antigen in Cre-induced Kras-mut/p53-KO lung tumors (cf DuPage et al Cancer Cell 2011 from Tyler Jacks group, which allows to study time-dependent immunoediting).

Fig.7B: TCGA data for lung cancer is strongly biased towards early stage disease (availability of sufficient material in resectable vs advanced lung cancer). As correlation is affected by sample size, this undermines the analysis performed and the conclusions drawn.

Fig. 7C: Waterfall plot: the y-axis label "TIM4 changes from baseline" is somewhat confusing from a clinical point of view as it could be interpreted as change during the course of treatment (eg during/after immunotherapy compared to baseline value before treatment start). I understand that RNAseq data was only obtained from samples pre-treatment? This should be more clearly stated, Also, the way the waterfall plot format is used is somewhat contrived. In order to indicate the

potential predictive value of a biomarker (which the authors claim TIM4 might be), usually best objective response is plotted per patient according to RECIST (% change in tumor load relative to baseline on y-axis), and bars are color-coded for biomarker-positive or negative status (eg above or below median). This would show more convincingly whether TIM4-positivity enriches for responders (even though this is a very small clinical dataset of n=16).

As patient survival data is available from TCGA, the authors should perform a Kaplan-Meier survival statistic of NSCLC patients with TIM4 expression e.g. above or below median. If indeed TIM4 ultimately promotes immune surveillance by allowing cross-priming of cytotoxic T-cells, then this should translate in a long-term prognostic impact of early stage NSCLC (as half of these patients experience disease relapse after resection due to the presence of remaining micro-metastases, which are potentially prone to control by the immune system). Prognostic in the sense of irregardless of treatment, which is more in line with the mouse data (in which the impact of TIM4 function on response to immune checkpoint blockade was not investigated).

MINOR COMMENTS:

"...Instead cDC1 showed similar engulfment in controls and early tumor- bearing lungs, whereas the uptake was significantly reduced in late tumor bearing lungs (Fig 1A)."

 Could it be that the antigen is unaccessible due to anatomical obstacle? Is there any idea in which pulmonary tissue compartment cDC1 reside (peribronchial? inter-alveolar?)? Could the presence of tumor just simply wall off access of cDC1 to intratracheally delivered apoptotic cells?

"... Cell-associated antigens were efficiently cross-presented inducing proliferation of OVA specific T cells in lung draining lymph nodes, whereas proliferation was substantially impaired in tumor draining LNs (Fig 1G). "

 is the migration of cDC1 to MLN suppressed?

"...When transferred into late tumors, OVA specific CD8+ T cells proliferated poorly (Fig 2 E,F), suggesting ineffective priming by antigen presenting cells in vivo. "

 could also very likely be due to accumulation of MDSCs, which is seen in orthotopic transplantable lung tumor models

"...immunogenic chemotherapy, which is based on release of neoantigens by cell death "

 immunogenic cell death is primarily based on the exposure of specific molecules such as calreticulin or DAMPs such as HMGB4 and ATP from dying cells, rather than availability of antigens

Methods section, Statistics: no description of correlation analysis shown in Fig. 7 A & B

Typos:

p.3: TIME  TME

p. 10: addictive  additive effect

Reviewer #1 (Remarks to the Author): with expertise in Tim4

In the manuscript, Caronni et al. show that cDC1 is a type of phagocytes engulfing tumor associated antigens, which is required for CD8+ T cell activation and proliferation preventing tumor progression in the KP lung tumor model. Especially, they show that Tim4, a PS receptor, exclusively is expressed in cDC1 and mediates engulfment of tumor associated antigen. In tumor progression, the level of Tim4 in cDC1 is downregulated, which causes less efficient engulfment of tumor associated fragments, resulting in less cross-presentation of tumor antigens by cDC1 to CD8+ T cells and thus decreased CD8+ T cell activation. Treatment of KP mice with an anti-Tim4 antibody or Tim-4 depleted mice show less engulfment of tumor cells by cDC1 and decreased cross-presentation and CD8+ T cell activation by cDC1. Overall, the expression of Tim4 in cDC1 and its effects on engulfment of tumor cells by cDC1 are solid and clearly show that Tim4-mediated engulfment of tumor cells by cDC1 is related to CD8+ T cell activation. There are however several issues that need to be addressed and the molecular mechanism by which Tim4 is downregulated in tumor progression needs to be established.

Major comments.

1. The significance of the study is the finding that Tim4 expressed in cDC1 mediates engulfment of lung tumor cells by cDC1, which is required for cross-presentation of tumor antigens by cDC1 to activate CD8+ T cells and that the level of Tim4 is downregulated in KP late. Although the lower level of Tim4 in KP late than in KP early is clear, the molecular mechanism by which Tim4 is downregulated in KP late is not addressed. Considering the novelty and findings of the study, how Tim4 is downregulated in KP late is an important question to be addressed.

R: The present study shows that cDC1 become unable to engulf and cross-present cell associated antigens in advanced lung tumors. Selective and high expression of the phosphatidylserine receptor TIM4 in normal lung cDC1 and its loss along tumor progression has been identified as the mechanism underlying inefficient phagocytosis. This contributes to demonstrate for the first time that crippling tumor antigen presentation by DC1 starts with inhibition of phagocytosis in tissue by modulation of phagocytic receptors.

We fully agree that the mechanism of downregulation, likely a complex combination of factors, is an important aspect, but we think this goes beyond the present scope.

Still we have performed some pilot experiments in an effort to comply with this request. Based on the IFN-I signatures in tumor associated DC1 (Fig S5B) we reasoned that IFN-I signalling may be involved. We thus challenged healthy controls with inducer of IFN-I (poli:IC and the STING agonist DMXAA) and recombinant IFN-I. The lungs were harvested after 2 days to measure TIM4 and MHC class-II. Poli:IC and IFN-I induced MHC class-II, indicating maturation whereas DMXAA did not induce maturation. Poli:IC substantially diminished TIM4 levels whereas DMXAA and IFN-I led to a slight increase.

These data suggest that triggering of innate responses may be involved in receptor modulation yet they are not conclusive therefore we restrain to publish them at this stage.

2.Regarding the comment above, it is unclear whether Tim4 downregulation in KP is causative for the tumor progression. In other words, it is unclear whether the tumor progression results in Tim4 downregulation or Tim4 downregulation causes the tumor progression. The authors may use Tim4 KO mice to monitor the tumor progression.

R: We agree with the reviewer on this suggestion however TIM4 full KO mice have some limitations given the impact of receptor deletion on abundant macrophages populations in the peritoneum, liver and spleen which may impact on the whole inflammatory environment. For this reason we opted for short receptor blockade by antibodies and we used full TIM4 KO mice only for short kinetics and local lung readouts. To study tumor growth we have specifically targeted TIM4 on cDC1 by generating mixed bone marrow chimeras from mixtures of $Batf3^{-/-}$ and $TIM4^{-/-}$ (or WT) bone marrows. Specific deletion of TIM4 in cDC1 reduced tumor specific CD8 responses and, most importantly, inhibited control of tumor growth (Fig 5 e-g). This result proves that TIM4 on cDC1 promotes antitumor immunity in early phases. We propose that factors secreted by progressing tumors (or tumor tissue) leads to TIM-4 downregulation, facilitating escape from immune surveillance as discussed at page 16 in the text.

3. Lung tumor cells seem to be mostly ingested by alveolar macrophages and engulfment of lung tumor cells by cDC1 is much inferior to that by the macrophages (based on the authors' data). Although cDC1 is major cells to cross-present tumor antigens, the authors have not checked the possibilities that cross-presentation by other APC, i.g. alveolar macrophages, could be affected during tumor progression. Especially, the authors show that genes involved in intracellular trafficking and antigen processing are modulated in cDC1. Thus, it is still possible that the similar modulation happens to alveolar macrophages, resulting in less efficient cross-presentation and the less number of T cell in KP late. In addition, population engulfing tumor cells

by cDC1 is less than 10% (about 6% for KP early and about 2.5 % for KP late, Fig 1A). Decreased engulfment by cDC1 in KP late is clear, but the basal engulfment capacity of cDC1 is very low. Could this change make the changes?

R: Consistently with previous literature, our data now displayed in revised Fig S3e show that tissue resident cDC1 efficiently cross-present tumor antigens whereas cDC2 and alveolar macrophages do not. In addition, as shown in Fig 2g,h priming in Batf3^{-/-} mice that lack DC1 is strongly inhibited. Thus we can rule out that decreased priming of CD8 T cells depends on defective cross-presentation by other APC.

As to the engulfment capacity of the various lung phagocytes, it is important to note that uptake early after delivery of apoptotic thymocytes (Fig 1a and S2a) is highest in macrophages, in line with previous literature (Desch AN, JEM 2011; Helft J, JCI 2012). However, when analyzing the uptake of fluorescent tumor cells by phagocytes in tumor tissues, the highest fluorescence is associated to DC1 (Fig 1b and S2b). This likely reflects the superior capacity of DC1 to maintain undigested phagocytic cargo, a well-established property of DC1 that accounts for the capacity to cross-present antigens (Salmon H, Cell 2016, Ioffe and Amigorena, Nat Rev Immunol 2012).

5. Once again, regarding to the comment above, the authors find that many genes, especially genes related to intracellular trafficking and antigen processing, are modulated in KP late. They focused on Tim4 without rationale excluding other modulated genes. The authors need to show whether trafficking or antigen processing is unaltered in KP late or to explain why they excludes trafficking and antigen-processing genes

We focused on TIM4 since this was the most compelling candidate to explain inhibition of phagocytosis described in Fig 1 since this receptor was found to be highly expressed in normal lung DC1 and highly down-regulated during tumor progression. Level of expression and modulation are plotted in a new graphical representation in revised Fig 3c, to reinforce this concept.

We haven't excluded that other changes may contribute to DC1 suppression, downstream of antigen uptake. Indeed, we show that tumor associated DC1 are impaired in antigen processing of OVA antigen given ex-vivo (Fig S3 G,H), but they produce normal cytokine levels (Fig S3 i,l). The goal here was to decipher what controls the initial steps of antigen acquisition upstream of antigen processing and presentation, since crippling of downstream antigen processing events is already described (discussion, page 15).

4. Tim4 is one of the best characterized phosphatidylserine (PS) receptor and mediates engulfment of apoptotic cells exposing PS on the cell surface. The authors adequately show engulfment of apoptotic thymocytes by cDC1, which is modulated by the level of Tim4 expressed in cDC1. However, it is unclear that engulfment of lung tumor cells by cDC1 occurs in a PS-dependent manner. The authors could

check whether PS is exposed on dying or dead lung tumor cells or tumor fragments and PS masking molecules such as V or a MFGE8 mutant recapitulate the phenotype of Tim4 KO or the anti-Tim4 antibody treatment.

This is an interesting point. In line with previous findings in other tumor models we have found that a consistent fraction of tumor cells growing in the lung stains positive for annexin V (revised Fig S7d). We also tried to block uptake by administering free annexin V directly in the lung to saturate PS. However we haven't observed any change on the fraction of engulfed BFP cells. We think this experiment is difficult to control for effective PS blockade and addressing this point would require mouse models that we had no access to during this time.

5.Usage of the blocking antibody against to Tim4 is a good strategy to evaluate the effects of Tim-4 on cross-presentation, T cell activation and so on. Besides this approach, because the authors have Tim4 KO mice, it could be more reliable to use Tim4 KO mice rather than the blocking antibody to validate the effects of Tim4. The blocking antibody was used in most of experiment to evaluate the effects of Tim4. It is strongly recommended to use Tim4 KO mice for validation of Tim4 effects on cross-presentation, CD8+ T cell proliferation, and experiments in Figure 6.

We absolutely agree with the reviewer that cross-presentation by TIM4 deficient lung DC1 is an important evidence to add to the findings obtained by receptor blockade. We have performed two experiments and the data reported in revised Fig 4g demonstrate that cross-presentation of tumor antigens by cell-sorted cDC1 from Tim4^{d^{-/-}} tumors is reduced (Revised Fig 4g).

Minor comments

The authors claim equal loading of tumor lungs with dying cells based on Fig S1G. Because no difference of engulfment by cDC2 or AM between KP early and late could result from other factors, this is inadequate

We agree and we corrected the sentence.

The percentage gated in the density map for KP BFP early in Figure 1B is too high to match the percentages in the bar graph in the figure.

This was a mistake and the representative plots were from a different experiment with different tumor burden, this has now been corrected to match the quantification, thanks for bringing it up.

Figure 1D. Assay for binding of apoptotic cells to phagocytes is usually performed by blocking internalization of apoptotic cells. The authors may use cytochalasin D or 4 oC incubation and need to show more cells (the images in the figure).

R: Cell sorted DC1 are too few and fragile to allow treatment with drugs as it is normally done with cultured cells. For binding we have incubated cell sorted DC with thymo for 30 minutes and we have scored cells attaching to the DC surface at the single cell level by confocal, thus not counting internalized cells (virtually zero at this time point). The cell density on a slide is very low, so higher magnification images are poorly informative. This is why we opted for representative zooms to illustrate the criteria used for the quantification plotted on the right.

The authors show the MFI in the histograms and percentages in the bar graphs. The author need to show the gating in the histograms to get the percentages in the bar graphs (1e, 1f, 2e, 2f, 4c, 4d, 4e, S5, S6).

R: We have indicated in histograms the gate used to calculate percentages.

Tim4 in the study is from mice. Thus, the correct notation is Tim4, Tim-4 or Timd4. In the text and figures, they are mixed.

R: We used TIM4 for protein, Timd4 for the mouse gene and TIMD4 for the human gene.

There is a typo, TIME in page 3. TME seems to be correct instead of TIME.

R: TIME is spelled in the intro and stands for tumor immune microenvironment.

Reviewer #2 (Remarks to the Author): with expertise in cDC1s and tumor immunology

In this manuscript, Caronni et al. identify a mechanism by which antigen presentation by conventional type 1 dendritic cells (cDC1s) is progressively inhibited during the course of tumor development in a model of Kras/p53 mutant lung adenocarcinoma. The authors provide evidence consistent with a model whereby loss of TIM-4 expression on cDC1s reduces their ability to uptake and cross-present tumor cell-associated antigens impairing, in turn, CD8+ T cell responses as tumors progress to late stages. This is an interesting finding of both fundamental and translational implications for the cancer immunology and immunotherapy field.

In terms of novelty, a previous study (Parekh et al. 2017) has already demonstrated that TIM-4 expression on cDC1s is required to take up apoptotic cells, although this study did not address this question in the context of early versus late stages of lung tumorigenesis. In contrast to the findings of the current manuscript, others have shown TIM-4 expression by myeloid cells is upregulated in tumors, and that TIM-4 reduces antigen presentation and inhibited tumor immunity triggered by chemotherapy-induced tumor cell death (Baghdadi et al. Immunity, 2013). Caronni et al. propose a different role for TIM-4 expression on myeloid cells, specifically cDC1s, in uptake of tumor

antigens and how its expression is modulated during tumor growth. Regarding the selectivity of expression of TIM-4 on cDC1s, others have already shown CD103+ cDC1s express higher TIM-4 than other myeloid cells (de Mingo Pulido et al. Cancer Cell 2018), so this message is not particularly novel. The comments below outline further analysis of the existing data and suggestions of new experiments that could strengthen the manuscript.

General comment:

Despite previous reports suggested expression of TIM4 by DCs, a detailed analysis of TIM4 expression in properly identified DCs subsets (complete set of markers and gating strategies) across different tissues was never reported. Baghdadi et al, proposes that TIM4 induces degradation of tumor antigens by autophagy, thereby inhibiting immunity. However the study is focused on macrophages. Some experiments refer to tumor-associated dendritic cells, which are insufficiently defined as MHCII^{high}/CD11c⁺ (detailed gating strategy are not reported in methods) but do not make a point on cDC1, which are known to have a specialized antigen processing machinery. We discuss the context specific role of this receptor and the notion that a different antigen processing machinery in macrophages and DCs may explain disparate findings (addressed in page 15-16 discussion). Differently from what has been reported in breast tumors (De Mingo Pulido), we do not observe de novo expression in phagocytes and TAM infiltrating lung tumors (Fig 3g), suggesting the tissue specific regulation of the receptor.

A major claim of the paper is that TIM-4 expression is progressively repressed during tumor development, which is associated with a progressive decrease in the ability of cDC1 to uptake tumor cell-derived material. To substantiate this claim the authors should consider the following points:

• Show consistently ALL comparisons of normal lung vs early tumors vs late tumors rather than just comparing normal lung vs late tumors throughout manuscript (e.g. Fig 1C, Fig 1G, Fig 3A).

R: We agree on this point and we have performed new experiments to reinforce the causal link between TIM4 expression over time and antigen uptake/antigen cross presentation by DC1. The results presented in revised Fig 1C and 1G show that early tumors DC1, that still express TIM4 (Fig 3f), have an intact capacity to bind to and engulf apoptotic thymocytes and to induce T cell activation in response to antigen loaded apoptotic thymocytes.

Perform kinetics of TIM-4 downregulation across tumor progression (i.e. a time course of events, rather than just two arbitrary early and late timepoints).

R: In revised Fig S1A and text we provide the histo-pathological criteria to classify early and late tumors. As reported in Fig 2 a,b the two stages are related to active T cell response or functional suppression. Once established, these time points have been used to analyze changes in TIM4 and

functionality of the DC1 compartment. We do not expect to gain essential information by further splitting the tumor stages in more categories.

•Analyze differences in TIM-4 expression and cross-presentation capacity of ex-vivo cDC1s isolated from early and late tumors rather than nLung vs late tumors

R: Fig 1f shows cross-presentation capacity of DC1 isolated from early and late tumor DC1.

• Perform comparisons between early and late tumors in an independent model (i.e. 3LL model).

R: We have added a time course of TIM4 expression in 3LL associated DC1, new data are presented in revised Fig S7C.

To substantiate the claim that TIM-4 plays a selective role in cDC1 the authors should show the effect of TIM-4 genetic ablation or Ab-mediated blockade in the acquisition of cell-associated material (uptake of apoptotic thymocytes or BFP-expressing KP cells) for ALL myeloid cell populations including monocytes and TAMs, not just cDC1s, cDC2 or alveolar macrophages. Related to this, could the authors change the representation of the uptake of material from BFP-expressing cells from the % of BFP+ over BFP+ CD45- cells to the % of BFP+ cDC1 over total cDC1 instead, which is the more appropriate quantification.

R: We would like to point out that TIM4 is only expressed by steady state cDC1 in the lung and not expressed by any other lung phagocytes (nor at steady state or tumors, 3 f,g), therefore we do not expect bystander effect of the blockade. In revised FigS7f we present the data on uptake of BFP by other myeloid cells upon TIM4 blockade to confirm that this is low and not affected by blockade.

In Fig 4a (TIM4 blockade or isotype on early tumors), data are expressed as % of BFP⁺cDC1/cDC1. In Fig 1b data are expressed as % of BFP⁺cDC1 over BFP CD45- cells to normalize for tumor burden that is very different in early versus late tumors.

The cross-presentation analysis of KP-OVA tumors in vivo and cDC1/CD8 co-cultures following TIM4 blockade should be repeated in Tim4d^{-/-} mice and chimeric mice with a specific TIM-4 deficiency in cDC1s to further substantiate the conclusion that TIM-4 blockade specifically limits cDC1 function rather than any other TIM-4 expressing cell that could also be targeted by the TIM-4 blocking antibodies. This would also exclude, among other possibilities, that the blocking Ab prevents uptake of tumor cell material in general. Similarly, the authors acknowledge that similarly to a general TIM-4 antibody blockade, TIM-4 deficiency in other cell subsets may account for the deficient CD8

responses in germline Tim4KO mice. The authors need to substantiate their data showing that the effect is largely on cDC1 and test if there are no other effects on non-cDC1 following Ab blockade. That similar defects are observed in Batf3^{-/-} does not prove this point as these mice have a general baseline defect in cDC1 and cross-presentation in general.

R: We agree that analysis of cross-presentation by TIM4 null cells is important to complement blocking studies. In revised Fig 4g we present data showing a reduction in cross-presentation by Tim4^{-/-} DC1 as compared to wt DC1, in agreement with data from TIM4 blockade (Fig 4f) and consistently with reduced priming of endogenous T cells in lungs of Tim4d^{-/-} mice (Fig 5c,d). As pointed out in reply to the previous comment, effect of the blockade of other lung phagocytes can be excluded. Moreover, we can quite confidently exclude any effect on macrophages in other district since tumors are confined to the lung at the early time points of our analyses. Indeed, the strongest evidence of specificity and demonstration of the importance of TIM4 selectively in cDC1 is provided by genetic deletion of TIM4 selectively in cDC1 (Fig 5 e,g).

The authors should consider the following points to strengthen the characterization of TIM-4 expression across the manuscript.

- **Regarding the exclusivity of TIM-4 expression on cDC1s, the authors look at most phagocytes including alveolar and interstitial macrophages but not TAMs (Figure 3E). In the profiling across multiple mouse tissues (Fig S5B,C) it is clear that macrophages expressed the highest TIM-4 across all tissue sites. Could the authors comment on this and show the expression levels of Timd4 transcripts in all populations analyzed by scRNAseq (Figure S5A) rather than only in the ‘DC/macrophage’ cluster?**
- **Given the expression of TIM-4 is not bimodal and thus defining positive and negative populations is challenging, the authors should also plot the MFI of TIM-4 (e.g. Figure 3C) rather than just the % of the parental population to facilitate comparisons between different settings (i.e. cDC1 from resting, early and late tumors). In this regard, Tim4d^{-/-} cells may be a more appropriate control for Flow Cytometry rather than FMO controls (e.g. Figure 3C), especially because typically auto-fluorescent cells show high levels of TIM4 expression (e.g. macrophages in Fig S5B).**
- **Likewise, it would be useful to see the expression levels of TIM-4 across multiple populations in the tumor and systemically in chimeric mice to validate the system, especially because FigS7C seems to show TIM-4 expression is about 2 fold lower in peritoneal macrophages in the Batf3^{-/-}:Tim4^{-/-} chimera relative to the WT:Tim4^{-/-} control**

R: We agree on the need to improve characterization of TIM4 expression and we thank the reviewer for this suggestion. We have deeply revised Figure 3 and S6 to show: 1) expression across all myeloid cells including TAM (revised

3g); 2) data from Tabula Muris are now in primary Figure 3 and show TIM4 expression in all lung cells (3d); 3) a second dataset of scRNA seq of MHCII^{high}, CD11c⁺ cells showing preferential expression of Timd4 transcripts in cDC1 and loss in KP tumors (Maier et al, Nature 2020); 4) staining controls using Timd4^{-/-} cells to show specificity of staining (Fig S6a); 5) TIM4 protein expression by flow expressed as MFI and % (revised 3f).

Concerning the high expression by macrophages this is well-established and not surprising (De Shepper, S., Cell 2018; Ghosn EE, PNAS 2019; Etzerodt a, JEM 2020), and it was used as control. Instead, expression by bona fide DCs subsets in different organs and tissues was less accurately documented. Importantly, the selective high expression in lung DC1 was never reported. We added a paragraph to better comment this aspect, as suggested.

Additionally, the fact that peritoneal macrophages in Tim4^{-/-} Batf3^{-/-} chimeras express less TIM4 relative to the WT:Tim4^{-/-} control is expected since the population is reconstituted by half of Timd4 ko bone marrow. Instead DC1 are only reconstituted by the marrow of Tim4^{-/-}, leading to a selective full deficiency in the cDC1 compartments.

Regarding the experiments administering TIM-4 blockade before chemotherapy, it is not possible to say whether this is downstream of TIM-4 blockade on cDC1s or other cells.

• **The authors could use the Timd4^{-/-}:Batf3^{-/-} bone marrow chimeras in the context of the chemotherapy experiments to better address this question, particularly given that others have shown opposite phenotype of TIM-4 in repressing tumor immunity triggered by chemotherapy (Baghdadi et al. Immunity, 2013).**

R: In this experiment we have restricted TIM4 blockade to two treatments just before administration of the drug. These conditions and the specificity of TIM4 expression in lung DC1 should guarantee that blockade in other organs does not impact on the growth of lung tumors. We consider that repeating it in chimeric mice is a long-term and complex experiment that will no add essential info to the data already presented in Fig 6.

The discrepancy with findings from Baghdadi may be again related to the different tumor model (we do not see expression of TIM4 by myeloid infiltrating tumor cells in the lung) and the focus on macrophages (chimeric mice are reconstituted with LysM-driven deficiency). We have discussed these discrepancies at page 15-16 of the revised manuscript.

• **What is the number and phenotype of cDC1 following TIM-4 blockade and are they specifically targeted?**

R: We have added a new panel in revised Fig S7e that shows effective blocking of the receptor, no changes in MHC class II molecules and no changes in numbers for DC1. The absolute numbers and MFI of DC2, mono and AMO is also displayed to document the effect of the blockade.

The mechanism of downregulation of TIM-4 in late but not early tumors is an interesting and potentially novel angle of the manuscript which is not addressed. Although identification of a mechanism is perhaps beyond the scope of this manuscript, the previous paper by Parekh et al. has shown that Vps34 is responsible for TIM-4 downregulation in cDC1s. So Vps34 could be a candidate to test and at least the authors should expand their discussion to integrate these previous findings.

As already discussed in reply to rev1, we acknowledge the importance of the mechanism of TIM4 downregulation in tumors and we will continue to investigate it in future studies. Vps34 is an interesting hypothesis yet we haven't observed any significant modulation of this trafficking protein in tumor-associated lung DC1 in our transcriptomic analysis (Table 1).

The human evidence in this study is somewhat conflicting and necessitates further work. The authors claim TIM-4 correlates with cDC1s in human malignancies, yet the major claim of the murine experiments is that TIM-4 expression on cDC1s is in fact downregulated in advanced cancers. This correlation is higher in stage 1 patients than in all stages, but there is still a decent positive correlation between TIM-4 expression and DC1s in all stages group.

- **The authors could analyze available lung cancer patients scRNAseq datasets to assess specificity of TIM-4 expression on cDC1s (e.g. Lambrechts et al. 2018 Nat Med).**
- **The authors could extend their analysis of the association of TIM-4 expression with outcomes from immunotherapy in other available datasets of other tumor types. If there is no association with response, this would be consistent with the author's claims that TIM-4 plays a specific role in lung cDC1s. Yet, given TIM-4 expression was shown to associated with cDC1 and CD8 T cell signatures it is likely that it will be associated with survival and response more broadly.**
- **What is the rationale for showing the association of TIM-4 with GzmK and not GzmB, A, perforin or any other mediator or effector molecule of cytotoxic immunity in Figure 7?**
- **The authors could test if TIM-4 is progressively downregulated in lung cancer as they imply by monitoring Timd4 transcript levels across different lung cancer stages in TCGA.**

R: In response to this comment we would like to point out that we cannot expect the same full loss of receptor expression in advanced human lung tumors that we have observed in homogenous late murine tumors. We have now modified the way of plotting the data showing correlation in stage-I against stage II-III and IV (as opposed to all stages) to better emphasize the differences in correlation values along progression (revised Fig 7b and Table2). We agree that this is not a full loss of correlation but we think it is sufficient to indicate that there is a reduction along progression in human lung adenocarcinomas.

As suggested, we have analysed scRNA expression data in the study by Lambrecht and in a second recent study by Maier (PMID 32269339, Nature 2020). Unfortunately, TIMD4 transcripts were detected only in very few myeloid cells, precluding any conclusion (see below).

Data from Lambrechts D. PMID: 29988129
scRNA seq on total lung cells

Only 54 out of 9756 myeloid cells express TIMD4

Data From Meier B. PMID: 32269339.
scRNA seq on CD11c+MHC-II high cells

Only 13 cells out of 6505 express TIMD4

We now show the correlation between TIM4 transcripts and the gene signature corresponding to effector CD8 T cells (including perforin and GZMKB described in Table2) in stage-I patients (revised 7a,b).

In parallel, we have collected a new evidence to support the anti-tumoral role of TIM4 in human tumors. By including TIM4 to the cDC1 signature for stratification of patients we found a positive association with survival that was not observed using only the cDC1 signature (revised 7c). This mild positive association with better prognosis is then emphasized under selective pressure of immunotherapy, as shown in original Fig 7d.

Collectively, we consider that these data complement mouse data and provide an initial indication that TIM4 expression may be linked to beneficial outcome in human lung cancers. These initial findings will be expanded to fully elucidate the role of TIM4 in human diseases, including flow cytometry, IHC, emerging scRNA data sets and analysis of other tumor types and immunotherapy treated cohorts in future studies.

Minor comments

Clarification is required on figure legend Figure 1E: “Data are from 2 independent experiments with 4 mice/group. Each dot represents the pool of 2 animals”.

This has been corrected, thanks.

Figure 1E: The authors need to provide further evidence and controls to prove the MHC-I-SIINFEKL complex staining is specific (e.g. is the

staining negative in other stained cells?

R: Although the antibody has been largely used and validated we agree that this is an important control to show. In revised Fig S2d we show labeling for cDC1, cDC2 and AM in OVA tumor bearing tissues against control nLung (n antigen expression). The results show a strong labeling exclusively in cDC1 in OVA tumor, confirming the superior cross-presenting ability of DC1 and the specificity of the staining.

- Figure 1F, 2E,F and 4F: Could the authors use a consistent method for assessing the proliferation of OT-I cells? What was the rationale for representing the data as a % of cells that divided at least once in some experiments or twice in others?

R: We quantified cells that underwent at least 1 cycle or 2 cycles depending on the overall extent of proliferation that in turn depends on antigen load. In Fig 4e and 1g proliferation following delivery of OVA apoptotic thymocytes is less.

- Figure 4C: quantification of pHRodo fluorescence should be done using the MFI as this is not a bimodal distribution.

R: MFI is indicated in the plot of Fig 4C.

- Gating on cDCs in the lung based on CD11c and MHCII expression is not enough to identify cDCs, especially cDC2s. As such their increase (Fig 1 SE) in cDC2s could reflect an increase in some sort of myeloid cells (not cDC2).

R: Correct. We have actually included CD24 in most experiments to discriminate DC2 from Mo-DC and this is now shown in the revised gating strategy (revised S1). We recalculated the frequencies based on the new gating. We do not have absolute numbers for a sufficient number of replicates as those quantifications come from initial exp where CD24 was not included and thus we labeled the population as CD11b DCs (revised S1e)

- “Genes involved in intracellular trafficking and antigen processing were modulated in tumor-associated cDC1 which may explain defective cross-presentation”. Which genes from those shown are involved in intracellular trafficking? Also, genes for antigen presentation are enriched in tumor cDC1s, in contrast with their finding that late tumour cDC1s fail to cross-present antigens. This point should be noted by the authors.

R: Manual annotation of some of the most significantly modulated genes (FigS5D) shows association with intracellular trafficking functions (Rab24, Vps28, Scamp1 etc). Yet it is true that others may instead promote cross-presentation so we have rephrased the sentence to include the possible dual significance of gene modulation.

Figure 7B: please include the p value or indicate which of the correlations is statistically significant.

R: The p values are reported in revised Table 2.

Figure S2 is representative of the late time point d28. What is the picture at the early time point

R: Cytokine production and cross-presentation of ex-vivo administered soluble antigens was added to complete characterization of the suppression phase, beyond our focus on uptake (now revised FigS3).). Adding these accessory data also for the early time point would require a large number of animals in spite of a relatively minor contribution to the message, so we have decided not to perform them. Early time points have been extensively analyzed for the key readouts of interest showing that functionality of DC1 is still intact (Fig 1 and 2).

- Spelling throughout manuscript: tymphocytes, interferon alfa pathway, Oxaliplatino, Dexamethason and grammar: Page 19: were been passed, were anticipated to day 5 or 7.

This has been done thanks

Reviewer #3 (Remarks to the Author): with expertise in lung cancer - mouse models and immunology

GENERAL ASSESSMENT

This paper investigates the role of TIM4, a receptor known to mediate uptake of apoptotic bodies by macrophages. The data indicates that lung cDC1, already known to be important in anti-tumor adaptive immunity, use TIM4 in a non-redundant manner for uptake of lung tumor-associated antigens and cross-priming of anti-tumoral CD8 T-cells. This phenomenon appears to be lost as locoregional tumor load increases. In a translational part, it is suggested that TIM4 is a theranostic biomarker of durable response to PD1 blockade in NSCLC. As such the paper provides incremental insights on the role of TIM4 in adaptive immunity to tumor-associated antigens, without bringing fundamental change in thinking within the field of tumor immunology.

MAJOR COMMENTS:

The paper could have been stronger with more extensive use of the autochthonous tumor model to assess this underexplored aspect of tumor-induced immune dysfunction, which is the temporal dimension and the evolution of immune dysfunction as tumor load increases in a continuous fashion. However, as far as I understand, for most mechanistic experiments the authors revert to adoptive transfer of tumor cells expressing a model antigen. The conclusions drawn from

the use of such “fast” transplantable models do not necessarily apply to more real-life spontaneous tumor models, e.g. when it comes to investigating immunoediting. With respect to this, the term “KP tumor lungs” as used throughout this report is confusing as it does not provide the reader with a distinction between lungs after adoptive transfer of KP cells (i.v. or intracostal), versus lungs with AdenoCre-induced tumors. See for example legend of Fig.1: what is exactly meant by “lungs bearing KP tumors”? Is the Cre-induced autochthonous tumor model only used in Fig S6B? Admittedly not all experiments can be performed in the autochthonous model (e.g. experiments involving genetic knockdown of Tim4^{-/-}), still there are models for instance that feature expression of a model antigen in Cre-induced Kras-mut/p53-KO lung tumors (cf DuPage et al Cancer Cell 2011 from Tyler Jacks group, which allows to study time-dependent immunoediting).

R: We agree with the reviewer on the limitations inherent to the use of transplantable tumor models as opposed to the autochthonous model. Yet we feel that KP derived lines with defined genetic mutations implanted in the original organ (as opposed to ectopic implantation) can still provide important mechanistic preclinical insights as demonstrated by many recent reports (Engblom, Science 2017; Binnewies, Cell 2019; Maier, Nature 2020). Additionally, we have validated TIM4 loss in the autochthonous model (Fig S7B) where the model is indicated as KP-GEMM. The definition of KP tumors as transplantable tumor is now more clearly introduced in the first paragraph of results to avoid confusion.

Fig.7B: TCGA data for lung cancer is strongly biased towards early stage disease (availability of sufficient material in resectable vs advanced lung cancer). As correlation is affected by sample size, this undermines the analysis performed and the conclusions drawn.

We agree on this point. We have now plotted correlations between TIM4 transcripts and gene signatures for DC1, effector T cells and other immune subsets for stage I versus stages II, III, IV (revised 7a,b). This emphasizes the decrease in correlation values in progressing tumor, suggesting loss of cDC1 and loss of TIM4 in advanced stages consistently with what has been observed in the mouse system and already reported in human lung tumors (Lavin et al, PMID: 28475900).

Fig. 7C: Waterfall plot: the y-axis label “TIM4 changes from baseline” is somewhat confusing from a clinical point of view as it could be interpreted as change during the course of treatment (eg during/after immunotherapy compared to baseline value before treatment start). I understand that RNAseq data was only obtained from samples pre-treatment? This should be more clearly stated,

Also, the way the waterfall plot format is used is somewhat contrived. In

order to indicate the potential predictive value of a biomarker (which the authors claim TIM4 might be), usually best objective response is plotted per patient according to RECIST (% change in tumor load relative to baseline on y-axis), and bars are color-coded for biomarker-positive or negative status (eg above or below median). This would show more convincingly whether TIM4-positivity enriches for responders (even though this is a very small clinical dataset of n=16).

R: We agree that the original depiction of data was not intuitive and we thank the reviewer for bringing it up. Indeed RNAseq was obtained before treatment. The patients of the study were classified into responders and not responders to PD-1 based on RECIST (PMID: 32111241). In the waterfall plot, we stratified the two groups based on TIM4 expression above or below median from RNAseq obtained before treatment initiation. We have now better specified this in the text and legend and we refer to TIM4 levels on the y axis of the waterfall plot.

Unfortunately we haven't found other accessible sequencing dataset from PD-1 treated patients to expand the (most are nanostring-based and do not include TIM4).

As patient survival data is available from TCGA, the authors should perform a Kaplan-Meier survival statistic of NSCLC patients with TIM4 expression e.g. above or below median. If indeed TIM4 ultimately promotes immune surveillance by allowing cross-priming of cytotoxic T-cells, then this should translate in a long-term prognostic impact of early stage NSCLC (as half of these patients experience disease relapse after resection due to the presence of remaining micro-metastases, which are potentially prone to control by the immune system). Prognostic in the sense of irregardless of treatment, which is more in line with the mouse data (in which the impact of TIM4 function on response to immune checkpoint blockade was not investigated).

R: We have performed the suggested analysis. TIM4 expression values above median by themselves are not sufficient to predict better survival. However, by stratifying TCGA samples for low quantile expression of TIM4 and cDC1 genes we observed a significant impact on survival that was not observed by using only cDC1 genes without TIM4. This may indicate that TIM4 can improve stratification for extreme cases (Fig 7c).

MINOR COMMENTS:

"...Instead cDC1 showed similar engulfment in controls and early tumor-bearing lungs, whereas the uptake was significantly reduced in late tumor bearing lungs (Fig 1A)."

 Could it be that the antigen is unaccessible due to anatomical obstacle? Is there any idea in which pulmonary tissue compartment cDC1 reside (peribronchial? inter-alveolar?)? Could the presence of tumor just simply wall off access of cCD1 to intratracheally delivered

apoptotic cells?

R: This is a pertinent question. DC1 were shown to localized in the epithelium of bronchioles and alveoli and to be able can extrude dendrites in the alveolar lumen (M.Guilliams, Mucosal Immunology, 2013). The major evidence that lung tumor cDC1 are intrinsically unable to engulf dying cells comes from data in Fig 1c,d. In those experiments, DC1 isolated from lung tumors and fed with apoptotic cells ex-vivo showed a profound impairment in uptake as compared to DC1 resident in normal lungs. Yet we cannot exclude that other anatomical barriers come into play in vivo. We do have initial evidences that apoptotic cells reach the tumor area by tissue IF, but these data are too preliminary to be presented.

"... Cell-associated antigens were efficiently cross-presented inducing proliferation of OVA specific T cells in lung draining lymph nodes, whereas proliferation was substantially impaired in tumor draining LNs (Fig 1G). "

 is the migration of cDC1 to MLN suppressed?

R: It is true that we have not tracked arrival of antigen carrying DC1 to the lymph node but it has been shown quite convincingly by previous reports that cDC1 migrate efficiently to tumor draining LN (Robert E.V., Cancer Cell 2016; Salmon H, Immunity 2016). This occurs in a CCR7 dependent manner and we haven't observed any loss of CCR7 in tumor DC1 in our gene expression analysis. Moreover we show that cell sorted lung tumor DC1 fail to present antigens ex-vivo, mapping the defect to inefficient acquisition of antigens in tumor tissues.

"...When transferred into late tumors, OVA specific CD8+ T cells proliferated poorly (Fig 2 E,F), suggesting ineffective priming by antigen presenting cells in vivo. "

 could also very likely be due to accumulation of MDSCs, which is seen in orthotopic transplantable lung tumor models

R: We agree that this experiment is not sufficient to establish defective priming by APC in vivo. This notion is corroborated by experiments in Fig 1f that show defective priming by isolated DC1 ex-vivo. We have rephrased for clarity.

"...immunogenic chemotherapy, which is based on release of neoantigens by cell death "

 immunogenic cell death is primarily based on the exposure of specific molecules such as calreticulin or DAMPs such as HMGB4 and ATP from dying cells, rather than availability of antigens

R: We agree on this. The combination of immunogenic drugs used here was previously shown to turn KP cells into a hot tumor by recruiting CD8 T cells (Pfirschke C Immunity 2016). This is why we opted for this drug combination even if release of neoantigens is not demonstrated. We have reformulated the

sentence for clarity.

Methods section, Statistics: no description of correlation analysis shown in Fig. 7 A & B

R: We have added a new table (table 2) showing the statistic for 7A&B.

Typos:

p.3: TIME  TME

p. 10: addictive  additive effect

REVIEWER COMMENTS

Reviewer #1 (Remarks to the Author):

In the revised manuscript, Caronni et al. have addressed my critiques with new experiments and discussion of the points I have raised. Overall, the authors have done well and the manuscript has been improved. Specifically, 1) The question about molecular mechanisms by which Tim4 is downregulated has been validated. 2) The causative role of Tim-4 has been addressed. 3) The effects of macrophages or other APCs in KP are relatively well-discussed. 4) The exposure of PS on tumor cells was tested. 5) Instead of anti-Tim4 antibody treatment, the effects of Tim4 were additionally evaluated in some experiments using Tim4 depleted mice. However, based on their revision, I have a few more suggestions to improve the rigor of the manuscript.

1. Some regulation mechanisms for the level of Tim4 has been suggested (PMID: 32251403, PMID: 29150239). Klf2, 4 and/or MAPK could be modulated during tumor progression, which could be tested.

2. Although the authors pointed out the experimental hurdles of the assay for binding of apoptotic cells to cDC1, the images are not reliable. Based on the other data shown in the manuscript (e.g., Fig 1a and 1b), it seems that sufficient cDC1 cells could be sorted and collected. The authors use their conditions. However, if the authors want to show the images for engulfment or binding of apoptotic cells by phagocytes, the images should be improved (with more cells with low magnification).

3. Fig. 1b. the representative contour plot (middle) does not show as a circle in the graph. The authors need to be extremely careful for the usage of representatives.

4. Fig. S7d. The sentence for the legend should be corrected. The cells do not express Annexin V.

5. Tim4 is an alias of Timd4. The gene name is Timd4 (T cell immunoglobulin and mucin domain containing 4). The authors is recommended to maintain the consistency of usage of the gene and protein name. In the manuscript, TIM4 and TIMD4 are mixed. I recommend using either.

Reviewer #2 (Remarks to the Author):

In this newly revised manuscript the authors have addressed many of my major concerns and the main findings have been strengthened. I think the authors provide a compelling story in which most conclusions are well supported by the data. However, some comments and questions were not fully or properly addressed and I feel it is important to do so.

Major comments:

Based on the new data added I still do not think the authors provide sufficient evidence for their conclusion that TIM4 is progressively downregulated during tumor progression. For this, they need to show more than two time points as I previously suggested or check TIM4 expression among cDC1 in the autochthonous model. More importantly, to investigate this potential progressive loss in human lung cancer (TCGA), they should analyze TIM4 transcript expression or its correlation to a cDC1 signature in the four lung cancer stages independently rather than comparing stage I against all the others stages grouped. Otherwise, they are not really showing 'progressive loss'. The fact that the correlation between TIM4 and cDC1 is less pronounced does not constitute evidence that the TIM4 expression is downregulated in cDC1.

The new univariate survival analysis in Fig 7c needs to be adjusted by tumor stage. The presumable prognostic utility of adding TIM4 to a cDC1 signature could simply reflect the fact that, as suggested, TIM4 expression is higher in early stages and so expected to be positively associated with better outcome. Overall, I still feel the human analysis is preliminary, as acknowledged by the authors in their rebuttal letter, and the authors could consider not including this figure in the manuscript.

Representative gating for TAMs should be depicted in FigS1.

I still do not think that representing the % of BFP+ out of total tumor cells is the right way to monitor uptake by a specific cell population and I do not understand the authors' argument that this is depicted in this way to normalize by the total tumor content. If they want to normalize by the tumor burden, they should show both the % of BFP+ cDC1 out of total cDC1 and their total number per gram of tumor.

Figure 3d shows that there is Timd4 expression on a cluster of dendritic cells/macrophages which includes cDC1 but one cannot tell if Timd4 is only expressed by cDC1. The authors need to perform the analysis re-clustering this wide cluster of myeloid cells so that cDC1 can be distinguished from TAM, cDC2, etc and one can appreciate if indeed TIM4 expression is restricted to cDC1. From the analysis of Maier at al single cell RNA seq, based on what data the authors conclude that Timd4 expression is lost in "tumor conditions"?

For Fig 3g and Figure S6b, the authors need to show their gating strategy for identifying TAMs and IM. It is not clear why they do not include these myeloid populations for the analysis in Fig S1. As stated in my original comments, it is important to do so across the different readouts for TAMs, as these often very abundant tumor-infiltrating cells are highly phagocytic. Curiously, based on the new Fig 3g, TAMs do not express TIM4 while other tissue macrophages (Fig S6c) express high levels of it. As mentioned by the authors, this data is in marked contrast to what has been reported for breast cancer and melanoma. This is quite interesting and an important novel finding so it is essential to more comprehensively show the comparison side by side of cDC1 with TAMs (in Fig 1a,b, in Fig S6a, Fig S7b, c, e).

The authors conclude that myeloid cells in murine lung tumors do not express TIM4. But cDC1 are myeloid cells!

For consistency, the authors should include the total number of proliferating cells across all experiments with transferred OT-I cells as done in fig S7h.

Quantification of pHRodo fluorescence is still in % not MFI.

"Genes involved in intracellular trafficking and antigen processing showed some degree of modulation in tumor-associated cDC1 (Fig S5c,d)" Why does the heatmap show inflammation related genes?

There a few instances where the % in the gating do not match any of the dots on the quantification to the right e.g. Fig 1b, Fig 5b.

Figure S3 - Why do authors skip from "i" to "l" and leave out j and k?

Page 8 - "Gene expression analysis returned more than 40 transcripts differentially expressed" But their figure only shows 36?

Previously identified typos were not corrected. In fact, the manuscript could do with a careful proofreading as it contains a considerable number of typos and grammatical mistakes.

Reviewer #3 (Remarks to the Author):

Thank you for providing responses to my comments.

I feel there are areas which still require improvement and/or clarification. Please find hereunder feedback on the responses to my major comments:

"R: We agree with the reviewer on the limitations inherent to the use of transplantable tumor models as opposed to the autochthonous model.

Yet we feel that KP derived lines with defined genetic mutations implanted in the original organ (as opposed to ectopic implantation) can still provide important mechanistic preclinical insights as demonstrated by many recent reports (Engblom, Science 2017; Binnewies, Cell 2019; Maier, Nature 2020). Additionally, we have validated TIM4 loss in the autochthonous model (Fig S7B) where the model is indicated as KP-GEMM. The definition of KP tumors as transplantable tumor is now more clearly introduced in the first paragraph of results to avoid confusion."

What is still unclear to me in e.g. Fig S7A&B is: what tissue is actually processed and used in control vs tumor-bearing lung? Is the whole lung processed? When digesting whole lung one compares apples and oranges as non-injected lungs or KP mice not treated with adenoCre do not contain tumor nodules. Thus it is impossible to ascertain where the DCs are coming from: intratumoral, or peritumoral lung tissue (could have an impact on its own). Please clarify further. In general, the term "normal lung" should be clearly defined in the figure legends: PBS i.v.? PBS i.t.?, AdenoCre in WT mice?, or KP mice without AdenoCre? depending on experimental setup. When using the AdCre-inducible model, the relevant comparison for S7B should be looking at TIM4 signal in cDC1 in early vs late induced tumor nodules, rather than comparing tumor-free vs tumor-bearing lungs. Also (fig S7B), 85% TIM-4+ cDCs in controls vs 75% TIM-4+ cDC1 in tumor-bearing tissue: is this a biologically relevant "loss"? In my opinion, fig.S7A and B are not useful and only create confusion.

"Fig 7B: We agree on this point. We have now plotted correlations between TIM4 transcripts and gene signatures for DC1, effector T cells and other immune subsets for stage I versus stages II, III, IV (revised 7a,b). This emphasizes the decrease in correlation values in progressing tumor, suggesting loss of cDC1 and loss of TIM4 in advanced stages consistently with what has been observed in the mouse system and already reported in human lung tumors (Lavin et al, PMID: 28475900)."

 The correlation analysis cannot be performed. It is a statistical problem, not a reflection of a biological phenomenon (ie loss of cDC1 and/or TIM4 in advanced disease). Note that stage II and a subset of stage III patients are still considered early stage / resectable disease. What is the rationale of pooling those patients with stage IV patients? Fig. 7B should be removed. It is also in contradiction with the concept defended by 7D and 7E: in these figures, a relationship is suggested between expression of TIM4 or cDC1 and outcome on anti-PD1. However, anti-PD1-treated NSCLC patients is a totally different population, ie strictly stage IV (metastatic) disease, in contrast to TCGA patients.

"Fig 7C: R: We agree that the original depiction of data was not intuitive and we thank the reviewer for bringing it up. Indeed RNAseq was obtained before treatment. The patients of the study were classified into responders and not responders to PD-1 based on RECIST (PMID: 32111241). In the waterfall plot, we stratified the two groups based on TIM4 expression above or below median from RNAseq obtained before treatment initiation. We have now better specified this in the text and legend and we refer to TIM4 levels on the y axis of the waterfall plot."

 My point is that the waterfall plot is used in an unconventional way (bar height and direction normally denote best tumor response per RECIST, color code for biomarker high vs low status), and is actually redundant next to the scatter plot next to it. I would remove the waterfall plot.

"R: We have performed the suggested analysis. TIM4 expression values above median by themselves are not sufficient to predict better survival. However, by stratifying TCGA samples for low quantile expression of TIM4 and cDC1 genes we observed a significant impact on survival that was not observed by using only cDC1 genes without TIM4. This may indicate that TIM4 can improve stratification for extreme cases (Fig 7c)."

 Were the KM survival stats performed on the LUAD & LUSC TCGA dataset as a whole (i.e. all disease stages pooled)? Note: looking at OS and PFS for stage I, II, III and IV separately may be informative. To raise hypotheses on prognostic value of cDC1/TIM4 (ie a measure of spontaneous immunosurveillance efficiency regardless of post-surgical treatment), one should focus on early stage NSCLC. (again note that only stage IV patients are treated with anti-PD1!). The survival curves should explicitly mention the biomarker cut-off used, and the cut-off clearly explained in the results or legend text. Also there is a contradiction between what is mentioned in the revised results section and in the rebuttal, versus what is written in the figure legend text (data source: TCGA vs GSE126044 cohort?)

REVIEWER COMMENTS

We thank all reviewers for rapid and careful evaluation of the revised version. We have done our best to address remaining concerns, correct mistakes and balance some of the conclusions. Here is the detailed reply to all your comments.

Reviewer #1 (Remarks to the Author):

In the revised manuscript, Caronni et al. have addressed my critiques with new experiments and discussion of the points I have raised. Overall, the authors have done well and the manuscript has been improved. Specifically, 1) The question about molecular mechanisms by which Tim4 is downregulated has been validated. 2) The causative role of Tim-4 has been addressed. 3) The effects of macrophages or other APCs in KP are relatively well-discussed. 4) The exposure of PS on tumor cells was tested. 5) Instead of anti-Tim4 antibody treatment, the effects of Tim4 were additionally evaluated in some experiments using Tim4 depleted mice.

However, based on their revision, I have a few more suggestions to improve the rigor of the manuscript.

1. Some regulation mechanisms for the level of Tim4 has been suggested (PMID: 32251403, PMID: 29150239). Klf2, 4 and/or MAPK could be modulated during tumor progression, which could be tested.

R: We thank the reviewer for this suggestion. Indeed, data in PMID: 32251403 showing that TIM-4 is lost in the elderly and can be rescued by p38 blockade are really compelling and represent the first example of negative modulation of TIM-4 expression, similarly to our observation in tumors. We refer to this paper in the discussion. Data in PMID: 29150239 refer to tissue signals and transcription factors that imprint macrophage identity in the peritoneum. Yet, as differential gene expression (Table 1) in lung cDC1 does not show changes in Klf2, 4 or MAPK, we speculate that regulation of TIM4 in lung cDC1 may be different. We will add these considerations in the discussion.

2. Although the authors pointed out the experimental hurdles of the assay for binding of apoptotic cells to cDC1, the images are not reliable. Based on the other data shown in the manuscript (e.g., Fig 1a and 1b), it seems that sufficient cDC1 cells could be sorted and collected. The authors use their conditions. However, if the authors want to show the images for engulfment or binding of apoptotic cells by phagocytes, the images should be improved (with more cells with low magnification).

R: We have provided few extra representative examples of larger fields for binding and more representatives of single cells for binding and uptake to further document the phenotype and the criteria used for quantification (revised S2). It is not possible to obtain more informative fields at large magnification for these small and rare primary populations as we start from low numbers of sorted DC1 and lose more cells during incubation, washing, binding to slides, staining. We do acquire several cells scanning the full coverslip and acquire at high magnification to have enough details to score them and get quantitative data. Data in Fig 1a,b are flow cytometry data from total lung cell suspension on gated cDC1, not cell sorted cDC1, so conditions are different. Indeed, representative images for rare, cell sorted populations like this are always shown as zoom on individual cells (PMID: 32516589; PMID: 32269339), so we think this is a common approach.

3. Fig. 1b. the representative contour plot (middle) does not show as a circle in the graph. The authors need to be extremely careful for the usage of representatives.

R: This was because representative showed percentage of BFP+ cDC1 and the graph showed % normalized on tumor cells. We have now replotted these data adding new experiments. The representative and the quantification in the graph are both plotted as % of BFP+ cDC1 on cDC1.

4. Fig. S7d. The sentence for the legend should be corrected. The cells do not express Annexin V.

This has been corrected in the legend.

5. Tim4 is an alias of Timd4. The gene name is Timd4 (T cell immunoglobulin and mucin domain containing 4). The authors is recommended to maintain the consistency of usage of the gene and protein name. In the manuscript, TIM4 and TIMD4 are mixed. I recommend using either.

Thank you for this. We have now indicated TIM4 for the protein and *Timd4* for the gene.

Reviewer #2 (Remarks to the Author):

In this newly revised manuscript the authors have addressed many of my major concerns and the main findings have been strengthened. I think the authors provide a compelling story in which most conclusions are well supported by the data. However, some comments and questions were not fully or properly addressed and I feel it is important to do so.

Major comments:

Based on the new data added I still do not think the authors provide sufficient evidence for their conclusion that TIM4 is progressively downregulated during tumor progression. For this, they need to show more than two time points as I previously suggested or check TIM4 expression among cDC1 in the autochthonous model. More importantly, to investigate this potential progressive loss in human lung cancer (TCGA), they should analyze TIM4 transcript expression or its correlation to a cDC1 signature in the four lung cancer stages independently rather than comparing stage I against all the others stages grouped. Otherwise, they are not really showing 'progressive loss'. The fact that the correlation between TIM4 and cDC1 is less pronounced does not constitute evidence that the TIM4 expression is downregulated in cDC1.

R: We identified two time points along tumor progression, characterized by clearly identified parameters (tumor burden and functionality of the T cell responses). We used the 2 time points to show that TIM4 is slightly decreased in early tumors and significantly down regulated in established tumors. Adding additional intervals in this relatively short time lapse (28 days) would not add much the main concept in spite of several additional animals to be used. We now avoid referring to progressive loss and only refer to expression in early and late tumors.

The analysis in the autochthonous model was performed after 10 weeks from tumor challenge. Histologically this corresponds to an advanced early stage (now displayed in revised S7b). TIM4 levels showed a small but significant decrease, indicating a behavior similar to that of transplanted tumors. We did not have a chance to analyze later stages of the autochthonous model to ascertain further loss. The sentence has been rephrased into "A slight reduction in TIM4 was observed as well in early autochthonous KP tumors".

Regarding the human data, we agree that loss of correlation across stages is not sufficient to claim loss of TIM4 expression on cDC1. We revised the figure to only present the correlation in stage I and excluded Fig 1b to avoid confusion.

The new univariate survival analysis in Fig 7c needs to be adjusted by tumor stage. The presumable prognostic utility of adding TIM4 to a cDC1 signature could simply reflect the fact that, as suggested, TIM4 expression is higher in early stages and so expected to be positively associated with better outcome. Overall, I still feel the human analysis is preliminary, as acknowledged by the authors in their rebuttal letter, and the authors could consider not including this figure in the manuscript.

Revised Figure 7c now show the significant prognostic value of the cDC1/TIM4 signatures only for stage I patients. We think it is important to present the human data (and the PD-1 study), even if preliminary, as they indicate the potential interest of further exploring TIM4 in modulation of myeloid functions in human lung tumors.

Representative gating for TAMs should be depicted in FigS1.

We added gating strategy for TAM in revised S5

I still do not think that representing the % of BFP+ out of total tumor cells is the right way to monitor uptake by a specific cell population and I do not understand the authors' argument that this is depicted in this way to normalize by the total tumor content. If they want to normalize by the tumor burden, they should show both the % of BFP+ cDC1 out of total cDC1 and their total number per gram of tumor.

We agree with this concern. We performed additional experiments to reinforce the result. New data (representative and quantifications) are presented in revised Fig 1b and are plotted as % of BFP+ DC1 out of total cDC1. It is not feasible to quantify gram of tumor for multifocal lung tumors. Lung weight itself is not a reliable measure because of the perfusion procedure used for harvest.

Figure 3d shows that there is Timd4 expression on a cluster of dendritic cells/macrophages which includes cDC1 but one cannot tell if Timd4 is only expressed by cDC1. The authors need to perform the analysis re-clustering this wide cluster of myeloid cells so that cDC1 can be distinguished from TAM, cDC2, etc and one can appreciate if indeed TIM4 expression is restricted to cDC1. From the analysis of Maier at al single cell RNA seq, based on what data the authors conclude that Timd4 expression is lost in “tumor conditions”?

Tabula Muris data in Figure 3d contain only control lungs and do not contain TAMs. The cluster corresponding to dendritic cells/macrophages cannot be further re-clustered as it only contains 85 cells. We have shown co-expression of DC1 markers on the TIM4 expressing cells (now added in the figure) and corrected the text. For scRNA data from Maier we added the statistic showing a significant decrease of expression in KP tumors (number of cells and level of expression included in the statistical test), which was not included in the previous version.

For Fig 3g and Figure S6b, the authors need to show their gating strategy for identifying TAMs and IM. It is not clear why they do not include these myeloid populations for the analysis in Fig S1. As stated in my original comments, it is important to do so across the different readouts for TAMs, as these often very abundant tumor-infiltrating cells are highly phagocytic. Curiously, based on the new Fig 3g, TAMs do not express TIM4 while other tissue macrophages (Fig S6c) express high levels of it. As mentioned by the authors, this data is in marked contrast to what has been reported for breast cancer and melanoma. This is quite interesting and an important novel finding so it is essential to more comprehensively show the comparison side by side of cDC1 with TAMs (in Fig 1a,b, in Fig S6a, Fig S7b, c, e).

We have presented the gating strategy for IM and TAM in S5 to accompany data in Fig 3. We would like to kindly remind that the focus of this study is on phagocytosis by cDC1 for cross-presentation of tumor associated antigens and the discovery that TIM4 controls this process, in a lung tumor model. Our data clearly show that TIM4 it is not expressed by lung TAMs. While we fully share the curiosity and intriguingness of the peculiar expression of TIM4 across macrophages subsets as discussed (page 16, discussion), we do not find that expanding this paper (at this stage of revision) to a full characterization of what happens to TAM phagocytosis during tumor progression represent an essential or reasonable request.

The authors conclude that myeloid cells in murine lung tumors do not express TIM4. But cDC1 are myeloid cells! **Since expression is lost in tumor cDC1, the resultant is no expression in myeloid cells in lung tumors. This sentence has been corrected to avoid misunderstanding.**

For consistency, the authors should include the total number of proliferating cells across all experiments with transferred OT-I cells as done in fig S7h.

Unfortunately, we could not retrieve absolute numbers for the experiments with Batf3 ko.

Quantification of pHRodo fluorescence is still in % not MFI.

The figure has been revised and includes MFI.

“Genes involved in intracellular trafficking and antigen processing showed some degree of modulation in tumor-associated cDC1 (Fig S5c,d)” Why does the heatmap show inflammation related genes?

Fig S5 c-d shows two heatmaps, one for inflammatory (c) and one for trafficking genes (d).

There a few instances where the % in the gating do not match any of the dots on the quantification to the right e.g. Fig 1b, Fig 5b.

Fig 1b has new data and graph showing in both cases % of BFP+ cDC1, whereas the previous version showed % in representative and normalized data in the graph. For 5b there was a mistake in copying the numbers on the plot, which has been corrected.

Figure S3 - Why do authors skip from “i” to “l” and leave out j and k?

This has been corrected.

Page 8 – “Gene expression analysis returned more than 40 transcripts differentially expressed” But their figure only shows 36?

As shown in Table 1, some differentially regulated genes are not annotated. Only annotated genes were displayed.

Previously identified typos were not corrected. In fact, the manuscript could do with a careful proofreading as it contains a considerable number of typos and grammatical mistakes.

Reviewer #3 (Remarks to the Author):

Thank you for providing responses to my comments.

I feel there are areas which still require improvement and/or clarification. Please find hereunder feedback on the responses to my major comments:

"R: We agree with the reviewer on the limitations inherent to the use of transplantable tumor models as opposed to the autochthonous model.

Yet we feel that KP derived lines with defined genetic mutations implanted in the original organ (as opposed to ectopic implantation) can still provide important mechanistic preclinical insights as demonstrated by many recent reports (Engblom, Science 2017; Binnewies, Cell 2019; Maier, Nature 2020). Additionally, we have validated TIM4 loss in the autochthonous model (Fig S7B) where the model is indicated as KP-GEMM. The definition of KP tumors as transplantable tumor is now more clearly introduced in the first paragraph of results to avoid confusion."

What is still unclear to me in e.g. Fig S7A&B is: what tissue is actually processed and used in control vs tumor-bearing lung? Is the whole lung processed? When digesting whole lung one compares apples and oranges as non-injected lungs or KP mice not treated with adenoCre do not contain tumor nodules. Thus it is impossible to ascertain where the DCs are coming from: intratumoral, or peritumoral lung tissue (could have an impact on its own). Please clarify further. In general, the term “normal lung” should be clearly defined in the figure legends: PBS i.v.? PBS i.t.?, AdenoCre in WT mice?, or KP mice without AdenoCre? depending on experimental setup.

When using the AdCre-inducible model, the relevant comparison for S7B should be looking at TIM4 signal in cDC1 in early vs late induced tumor nodules, rather than comparing tumor-free vs tumor-bearing lungs. Also (fig S7B), 85% TIM-4+ cDCs in controls vs 75% TIM-4+ cDC1 in tumor-bearing tissue: is this a biologically relevant “loss”? In my opinion, fig.S7A and B are not useful and only create confusion.

R2: Whole lung tissues are processed, following the procedures used by other recent studies (Zillionis, Immunity 2019; Maier et al Nature 2020, just as recent examples), and healthy mice are used as controls (better explained in legend of Fig 1a). For technical reasons, it is very hard to reliably harvest single nodules and separate them from juxta region (and obtain sufficient DC1 numbers) for mouse lung tumors, which contain multiple small nodules.

Yet, analysis of all tissue resident DC1 reveals major changes, indicating that this is a valid approach to identify pathways of interest. For the autochthonous model, it is now better specified in the legend that KP mice not treated with adenoCre are used as controls. The analysis in the autochthonous model was performed after 10 weeks from tumor challenge. Histologically this corresponds to an advanced early stage (now displayed in revised S7b). TIM4 levels showed a small but significant decrease, indicating a behavior similar to that of transplanted tumors. We did not have a chance to analyze later stages of the autochthonous model to ascertain further loss. The sentence has been rephrased into “A slight reduction in TIM4 was observed as well in early autochthonous KP tumors”.

"Fig 7B: We agree on this point. We have now plotted correlations between TIM4 transcripts and gene signatures for DC1, effector T cells and other immune subsets for stage I versus stages II, III, IV (revised 7a,b). This emphasizes the decrease in correlation values in progressing tumor, suggesting loss of cDC1 and loss of TIM4 in advanced stages consistently with what has been observed in the mouse system and already reported in human lung tumors (Lavin et al, PMID: 28475900)."

 The correlation analysis cannot be performed. It is a statistical problem, not a reflection of a biological phenomenon (ie loss of cDC1 and/or TIM4 in advanced disease). Note that stage II and a subset of stage III patients are still considered early stage / resectable disease. What is the rationale of pooling those patients with stage IV patients? Fig. 7B should be removed. It is also in contradiction with the concept defended by 7D and 7E: in these figures, a relationship is suggested between expression of TIM4 or cDC1 and outcome on anti-PD1. However, anti-PD1-treated NSCLC patients is a totally different population, ie strictly stage IV (metastatic) disease, in contrast to TCGA patients.

R: We agree on this comment and we have removed Fig 7b. We analysed recent scRNA seq data to make a firm point on loss of TIM4 expression in cDC1 in advanced human lung tumors but unfortunately there are not enough cells expressing Timd4 transcripts to draw a conclusion. We modified the text and specify that these analyses are a preliminary indication of co-expression of TIM4 by DC1.

"Fig 7C: R: We agree that the original depiction of data was not intuitive and we thank the reviewer for bringing it up. Indeed RNAseq was obtained before treatment. The patients of the study were classified into responders and not responders to PD-1 based on RECIST (PMID: 32111241). In the waterfall plot, we stratified the two groups based on TIM4 expression above or below median from RNAseq obtained before treatment initiation. We have now better specified this in the text and legend and we refer to TIM4 levels on the y axis of the waterfall plot."

 My point is that the waterfall plot is used in an unconventional way (bar height and direction normally denote best tumor response per RECIST, color code for biomarker high vs low status), and is actually redundant next to the scatter plot next to it. I would remove the waterfall plot.

Ok, suggestion taken.

"R: We have performed the suggested analysis. TIM4 expression values above median by themselves are not sufficient to predict better survival. However, by stratifying TCGA samples for low quantile expression of TIM4 and cDC1 genes we observed a significant impact on survival that was not observed by using only cDC1 genes without TIM4. This may indicate that TIM4 can improve stratification for extreme cases (Fig 7c)."

 **Were the KM survival stats** performed on the LUAD & LUSC TCGA dataset as a whole (i.e. all disease stages pooled)? Note: looking at OS and PFS for stage I, II, III and IV separately may be informative. To raise hypotheses on prognostic value of cDC1/TIM4 (ie a measure of spontaneous immunosurveillance efficiency regardless of post-surgical treatment), one should focus on early stage NSCLC. (again note that only stage IV patients are treated with anti-PD1!). The survival curves should explicitly mention the biomarker cut-off used, and the cut-off clearly explained in the results or legend text. Also there is a contradiction between what is mentioned in the revised results section and in the rebuttal, versus what is written in the figure legend text (data source: TCGA vs GSE126044 cohort?)

The KO survival stats were performed on disease stages pooled. We agree on this point and we now plot only stage-I. The biomarker cut-off are now explained in the legend and methods.

REVIEWERS' COMMENTS

Reviewer #2 (Remarks to the Author):

All my comments have now been satisfactorily addressed in this revised version of the manuscript.

Reviewer #3 (Remarks to the Author):

The responses address my latest comments for the most part.

I still have 2 issues to raise.

A minor issue is the fact that "normal lung" is still not clarified in fig 1. I understand now more clearly it is PBS i.v. mice as the figure refers to a lung-targeted transplantable tumor model via i.v. route.

This points to a less minor issue, explaining some of the confusion arising around the tumor model in this paper. In the introduction, the authors write: "... during development of Kras/p53 lung adenocarcinomas (KP), a model that accurately recapitulates lesions in human Non-Small Cell Lung Cancer (NSCLC)."

 For anyone familiar in this field, this sentence suggests a paper that is largely if not entirely based on the autochthonous, AdCre-induced lung tumors. In reality, the paper is almost entirely based on a transplantable tumor model (using KP-derived cell line), which does not really recapitulate human NSCLC. The AdCre paper if I understand well is only present in suppl. fig. 7. I would rewrite this sentence in the intro more accurately, in its current form it biases and confuses the reading of the data produced.

Point by Point reply

Reviewer #3 (Remarks to the Author):

The responses address my latest comments for the most part.
I still have 2 issues to raise.

A minor issue is the fact that "normal lung" is still not clarified in fig 1. I understand now more clearly it is PBS i.v. mice as the figure refers to a lung-targeted transplantable tumor model via i.v. route.

This points to a less minor issue, explaining some of the confusion arising around the tumor model in this paper. In the introduction, the authors write: "... during development of Kras/p53 lung adenocarcinomas (KP), a model that accurately recapitulates lesions in human Non-Small Cell Lung Cancer (NSCLC)."

 For anyone familiar in this field, this sentence suggests a paper that is largely if not entirely based on the autochthonous, AdCre-induced lung tumors. In reality, the paper is almost entirely based on a transplantable tumor model (using KP-derived cell line), which does not really recapitulate human NSCLC. The AdCre paper if I understand well is only present in suppl. fig. 7. I would rewrite this sentence in the intro more accurately, in its current form it biases and confuses the reading of the data produced.

R: We have now specified in the introduction, first paragraph of results and legend of Supplementary Fig 1a and Primary Figure 1a, in order to clarify what control were used along the study.